# Improved in vivo gene knockout with high specificity using multiplexed Cas12a sgRNAs

Fillip Port [1,2,3] ✉, Martha A. Buhmann[1,2,3,4], Jun Zhou [1,2,3,5], Mona Stricker[1,2,3], Alexander Vaughan-Brown [1,2,3], Ann-Christin Michalsen[1,2,3], Eva Roßmanith[1,2,3], Amélie Pöltl[1,2,3], Lena Großkurth[1,2,3], Julia Huber[1,2,3], Laura B. Menendez Kury[1,2,3], Bea Weberbauer[1,2,3], Maria Hübl[1,2,3], Elli Puscher[1,2,3], Florian Heigwer [1,2,3,6] & Michael Boutros [1,2,3] ✉

CRISPR nuclease-mediated gene knock-out is limited by suboptimal sgRNAs, inaccessible target sites, and undesired repair outcomes. Here, we present a Cas12a-based system in *Drosophila* that targets each gene with four sgRNAs to overcome these limitations. Multiplexed sgRNAs act through redundancy and synergism, frequently creating deletions between target sites and increasing the fraction of loss-of-function mutations. We show that multiplexed gene targeting is well tolerated and does not cause widespread proximity effects. To visualize CRISPR-nuclease activity in living animals, we developed a screening assay and used it to assess Cas12a activity across 33% of the *Drosophila* genome in combination with over 2000 sgRNAs. This revealed remarkably high on-target (>99%) and very low (<1%) off-target activity of multiplexed Cas12a sgRNA arrays. Quantitative side-by-side comparisons with current Cas9-based systems targeting over 100 genes in parallel demonstrate that multiplexed Cas12a gene targeting achieves superior performance and reveals phenotypes missed by established methods. The system described here provides a framework for reliable gene knock-out in multicellular systems.

CRISPR gene editing has significantly accelerated the functional annotation of genomes[1–3]. Despite its transformative impact, several factors limit the efficiency of gene disruption through CRISPR nucleases. For example, suboptimal activity of sgRNAs or Cas proteins can impede the induction of DNA breaks and chromatin state and genetic variation can render target sites inaccessible. Moreover, repair of DNA double strand breaks (DSBs) typically results in a spectrum of mutations that includes undesired repair outcomes retaining gene function, such as small in-frame insertions or deletions (indels). As a result, CRISPR editing in multicellular systems produces genetic mosaics - populations of cells carrying different mutations or

maintaining wild-type sequences. This severely compromises the sensitivity and robustness of functional gene discovery with CRISPR in animals, plants and cultured cells, presents a hurdle for the development of genetic control strategies for disease vectors, and limits its potential therapeutic application in cell and gene therapy.

A promising strategy to overcome these limitations is the simultaneous targeting of genes with multiple sgRNAs[4–6]. Here, additional sgRNAs can compensate for those that fail to mediate the desired outcome. However, for Cas9, the most widely used CRISPR nuclease, creating multiplexed arrays of more than two sgRNAs at scale remains technically challenging. The large size and repetitive nature of such

[1]Division of Signaling and Functional Genomics, German Cancer Research Center (DKFZ), Heidelberg, Germany. [2]Institute for Human Genetics, Medical Faculty Heidelberg, Heidelberg University, Heidelberg, Germany. [3]Department of Cell and Molecular Biology & BioQuant, Heidelberg University, Heidelberg, Germany. [4]Molecular Biosciences/Cancer Biology Program, Heidelberg University and German Cancer Research Center (DKFZ), Heidelberg, Germany. [5]Present address: The Affiliated XiangTan Central Hospital of Hunan University, School of Biomedical Sciences, Hunan University, Changsha 410082, China. [6]Present address: Department of Life Sciences and Engineering, University of Applied Sciences Bingen, Bingen am Rhein, Germany. ✉e-mail: f.port@dkfz.de; m.boutros@dkfz.de

arrays precludes them from being synthesized by commercial providers and manual assembly is time and cost intensive[4,6]. Most current large-scale Cas9 sgRNA libraries thus employ only one or two sgRNAs per gene[7–12]. In contrast, Cas12a nuclease is guided solely by small crRNAs (hereafter sgRNAs) and can autonomously process compact arrays into individual sgRNAs[13]. These properties allow Cas12a sgRNA arrays to be encoded on commercially available oligonucleotides, substantially facilitating generation of expression plasmids at scale. This has been harnessed for multiplex gene editing and transcriptome engineering in various contexts[5,14–17], but whether higher-order multiplexing of sgRNAs targeting the same gene enhances the efficiency of gene disruption in vivo remains unclear.

A key challenge of CRISPR gene editing is the occurrence of undesired on- and off-target mutations that obscure the phenotypic effects of disrupting the target gene. Large deletions, chromosome arm truncations and chromothripsis can arise from double strand breaks at the on-target site and compromise unrelated genes located in cis[18–22]. Such events have been demonstrated to occur with relevant frequencies in mammalian cells and some other organisms, but their relevance for gene targeting in *Drosophila* has so far not been investigated. In addition, off-target mutations can arise when Cas nucleases cleave distant sites with imperfect homology to the sgRNA[23,24]. Although computational tools can predict some off-target sites, their reliability is limited by poorly understood sgRNA-specific effects and sequence differences between reference and experimental genomes. However, methods for the unbiased, experimental detection of off-target effects in the genome of most animals are limited. Methods for the direct detection of induced DNA breaks have been described, but are not available for most species due to the absence of specialized tools like antibodies binding specific DSB repair proteins[25,26]. Whole genome sequencing is broadly available, but its application for comprehensive off-target analysis across many sgRNAs is limited by cost. Moreover, since CRISPR-induced mutations and natural genetic variation are often indistinguishable, inferring causal relationships about off-target effects from sequencing data alone remains non-trivial[27,28]. Consequently, the off-target activity of CRISPR systems in many organisms remains unclear, as many studies evaluate activity at only a very small selection of loci or with a small number of sgRNAs or do not test for off-target cleavage at all.

Here, we describe a collection of optimized, inducible Cas12a transgenes and quadruple sgRNA arrays for targeted mutagenesis of over 800 genes. Multiplexed sgRNAs provide redundancy and act synergistically to frequently induce deletions between target sites, generating mutations that are more likely to disrupt gene function. We performed large-scale activity screening and benchmarking against current Cas9-based technologies, which revealed very high on-target activity with more than 99% of lines being active, and demonstrated superior knockout performance across many target genes and tissues. We also systematically investigated potential undesired effects including toxicity, proximity effects, and off-target activity. These analyses revealed that multiplexed gene targeting is well tolerated and highly specific, with perturbations highly confined to the on-target locus.

## Results

### Synergistic action of quadruple Cas12a sgRNA arrays

We set out to implement and evaluate multiplexed Cas12a gene targeting in the tractable *Drosophila* model system. The short generation time of this organism enables comparisons across many conditions and replicates and the ability to encode nucleases and sgRNAs on single copy transgenes in defined genomic landing sites eliminates variations in delivery efficiency or expression level as confounding factors of experimental results. We previously showed that a variant of *Lachnospiraceae bacterium* Cas12a with a D156R mutation (hereafter Cas12a[+]) functions with high efficiency in *Drosophila* and can process

large sgRNA arrays[29]. To pave the way for the creation of systematic resources for multiplexed Cas12a[+] gene targeting, we first aimed to determine the optimal number of sgRNAs. We developed a reporter assay using genomic insertions of an exogenous sequence containing one to six target sites of a known, highly active sgRNA (Fig. 1a). This represents a controlled setup devoid of the typical variations in DNA cutting activity when different sgRNAs are used. Gene editing with ubiquitously expressed Cas12a[+] and single or multiplexed sgRNA target sites generated a variety of editing outcomes in somatic cells and the germline. Editing at single target sites primarily produced small indels, while targeting multiple sites progressively diversified the target locus and induced larger deletions between sites. This was evident from the migration pattern of amplicons from the target locus in somatic cells (Fig. 1b), which progressively transitioned to a 'smear' indicating heteroduplex DNA formation during PCR, and from sequencing reads of individual germline-transmitted alleles, which revealed frequent deletions between target sites (Fig. 1c). We selected a design with four sgRNAs per gene for in-depth characterization to balance the benefits of multiplexing against potential drawbacks, which could include excessive DNA damage, proximity effects, and off-target activity. This design choice is further supported by recent evidence that arrays consisting of more than 4 sgRNAs are no longer optimally processed by Cas12a[5].

Next, we evaluated the effects of multiplexed gene targeting on the integrity of eight endogenous loci. Here, different sgRNAs are used to target independent sites within each gene, which is expected to result in more diverse editing outcomes compared to the reporter system described above. PCR amplicons from genes targeted with quadruple sgRNA arrays and Cas12a[+] showed heterogeneous migration patterns during agarose gel electrophoresis (Fig. 1d). We observed a pronounced decrease in full-length amplicon abundance with a concomitant increase in alternative products for six target genes, an effect that was weaker for one gene and absent for another. In parallel, we targeted the same genes with lines of our previously described Cas9 sgRNA library, which uses two sgRNAs per gene[10]. Using this design, the reduction in full-length amplicons was substantially less pronounced, being strong for one, but weak or absent for the remaining seven target genes.

Together, these results suggest that Cas12a[+]-mediated targeting with quadruple sgRNA arrays produces greater target locus diversification compared to dual-sgRNA Cas9 editing. Notably, this comparison contrasts the system described here with the dual-sgRNA approach used in many publicly available *Drosophila* sgRNA lines and does not constitute a direct comparison of nuclease performance, which would require matched sgRNA designs. Collectively, these findings, together with published reports[4,30,31], suggest that sgRNA multiplexing could enhance gene disruption through two mechanisms (Fig. 1e). First, multiple sgRNAs provide functional redundancy, allowing active guides to compensate for inactive ones or those producing silent mutations. Second, multiplexed sgRNAs can act synergistically by generating simultaneous cuts that create deletions between target sites, which are more likely to yield loss-of-function alleles.

### Generation of a multiplexed Cas12a sgRNA library

To facilitate conditional gene knock-out in *Drosophila* with multiplexed sgRNAs and Cas12a[+] we created a large, publicly available toolbox of sgRNA and nuclease expressing fly strains. First, we designed a library of quadruple sgRNA arrays using our previously described CRISPR Library Designer[32]. sgRNAs were selected according to several criteria, including lack of predicted off-target sites, targeting of common exons, optimization of target site position and spacing, and target site-specific parameters (see methods). sgRNAs are expressed from the strong, ubiquitous U6:3 promoter and integrated at a defined attP landing site on the second chromosome (Fig. 2a). The

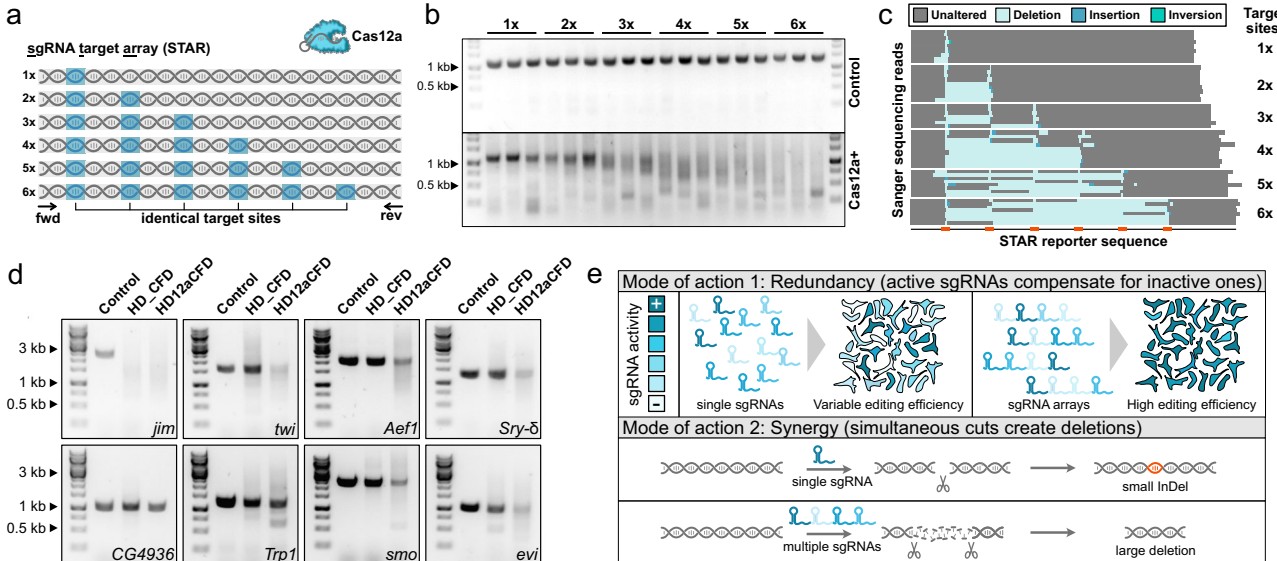

**Fig. 1 | Gene targeting with multiplexed sgRNAs leads to increased diversification of the target locus. a** sgRNA target array (STAR) reporter design for assessing multiplexed CRISPR editing outcomes. It consists of a non-coding sequence containing different numbers of the same sgRNA target site targeted by Cas12a+ and a single sgRNA in vivo. **b** Bulk editing outcomes assessed by PCR analysis of genomic DNA isolated from whole adult flies. In contrast to samples without nuclease (control; upper panel), targeting of the reporter results in amplicons of various sizes. Additional target sites produce more diverse editing outcomes, leading to increased diversity in gel migration behavior through formation of heteroduplex DNA and fewer amplicons of original length. Samples from three individual animals are shown for each genotype. **c** Each horizontal line represents the sequence of an individual STAR allele isolated through germline transmission and identified by sequencing. Multiplexing of target sites results in frequent deletions between target sites. **d** Bulk editing outcomes after mutagenesis of endogenous genes. The target locus was amplified from larval tissue by PCR

using identical conditions and amplicons were visualized by agarose gel electrophoresis. Control samples express sgRNAs targeting another locus. CRISPR mutagenesis frequently leads to a diversification of PCR amplicons. This effect is notably stronger when editing is performed with Cas12a+ and arrays of four sgRNAs per gene, as compared to Cas9 and two sgRNAs per gene, for 6 of the 8 tested target genes. Experiments reported in (**b**–**d**) used broadly expressed *act5c-Cas12a+* and *act5c-Cas9; tub-Gal4* (in d) and are representative of at least three independent replicates. **e** Two principal mechanisms of enhanced gene knockout using sgRNA arrays. Using several sgRNAs per gene in parallel improves knockout efficiency through redundancy, as active sgRNAs compensate for inactive ones. Knockout efficiency is further improved through synergistic action of sgRNAs, when simultaneous cuts can mediate deletions between target sites, which are more likely to constitute loss of function alleles than small indels. The schematic illustrates the effect on multiple cells, but the same effects apply when sgRNAs are delivered to entire organisms.

ability to encode all four sgRNAs on a single oligonucleotide allowed for synthesis and cloning in a pooled format, resulting in substantial savings in cost and time compared to the construction of our previously described Cas9 dual sgRNA library (see methods)[10]. So far, we have created Cas12a sgRNA lines targeting over 800 unique genes (Supplementary Data 1). We refer to these lines as the HD12aCFD sgRNA library. In parallel, we also created optimized Cas12a+ expression constructs (Fig. 2b). These feature a Gal4-dependent UAS promoter for tissue-specific expression, an upstream open reading frame to avoid excessive expression levels, and dual nuclear localization signals for efficient translocation into the nucleus (Fig. 2b, and Supplementary Fig. 1). To facilitate combination of these transgenes with different Gal4 drivers, we created independent insertions of the Cas12a+-expression plasmids on all three major chromosomes. We also constructed an additional line in which Cas12a+ expression can be controlled through excision of a FRT-flanked GFP cassette (Fig. 2b, and Supplementary Fig. 1).

Lastly, we generated a reporter that can be used to visualize CRISPR-Cas12a activity in vivo (Fig. 2c). Analogous to previously described reporters for Cas9 activity[33,34], this consists of an inhibitory upstream open reading frame in front of a GFP gene, which can be targeted by a specific Cas12a or Cas9 sgRNA. Mutagenesis of the reporter leads to the expression of GFP, which can be visualized in living animals or fixed tissues (Fig. 2d). Due to high activity and the permanent nature of the induced mutations, CRISPR mutagenesis can deviate from established Gal4 expression patterns (Fig. 2e-g)[10]. It is therefore advisable to empirically establish the activity pattern of a newly generated Gal4 UAS-Cas12a+ line.

## Multiplexed Cas12a gene targeting of single genes is well tolerated

Before exploring further the knockout efficiency of multiplexed Cas12a+ gene targeting, we investigated potential adverse effects of using several sgRNAs to disrupt the same gene. Since DNA DSBs are detrimental to cell fitness, one concern is that inducing multiple breaks simultaneously could overwhelm the DNA damage response, leading to excessive cell death[35–37]. To assess this risk, we quantified apoptosis levels following gene targeting with multiple sgRNAs by staining cells for activated effector caspase DCP-1. Induction of DSBs in non-essential genes caused a moderate increase in apoptosis compared to control tissue without DNA damage (Fig. 3a, b). Importantly, the level of cell death was not significantly different whether gene targeting was performed with Cas9 and two sgRNAs or Cas12a+ and four sgRNAs (Fig. 3a, b). This finding aligns with our functional validation experiments (see below), where multiplexed targeting of characterized genes using HD12aCFD yielded exclusively target-specific phenotypes. We also assessed toxicity after targeting genes present on multiple chromosomes as a result of their use as transgenic markers (Supplementary Fig. 2). We observed a marked increase in apoptotic cells in these genotypes. This was independent of the total number of sgRNA target sites, as a genotype with only four target sites scattered across the genome resulted in more cell death compared to eight sites clustered at a single locus (Fig. 3a, b, and Supplementary Fig. 2). These results indicate that while multiple DSBs in close proximity are well tolerated, parallel induction of DNA breaks on different chromosomes is highly toxic.

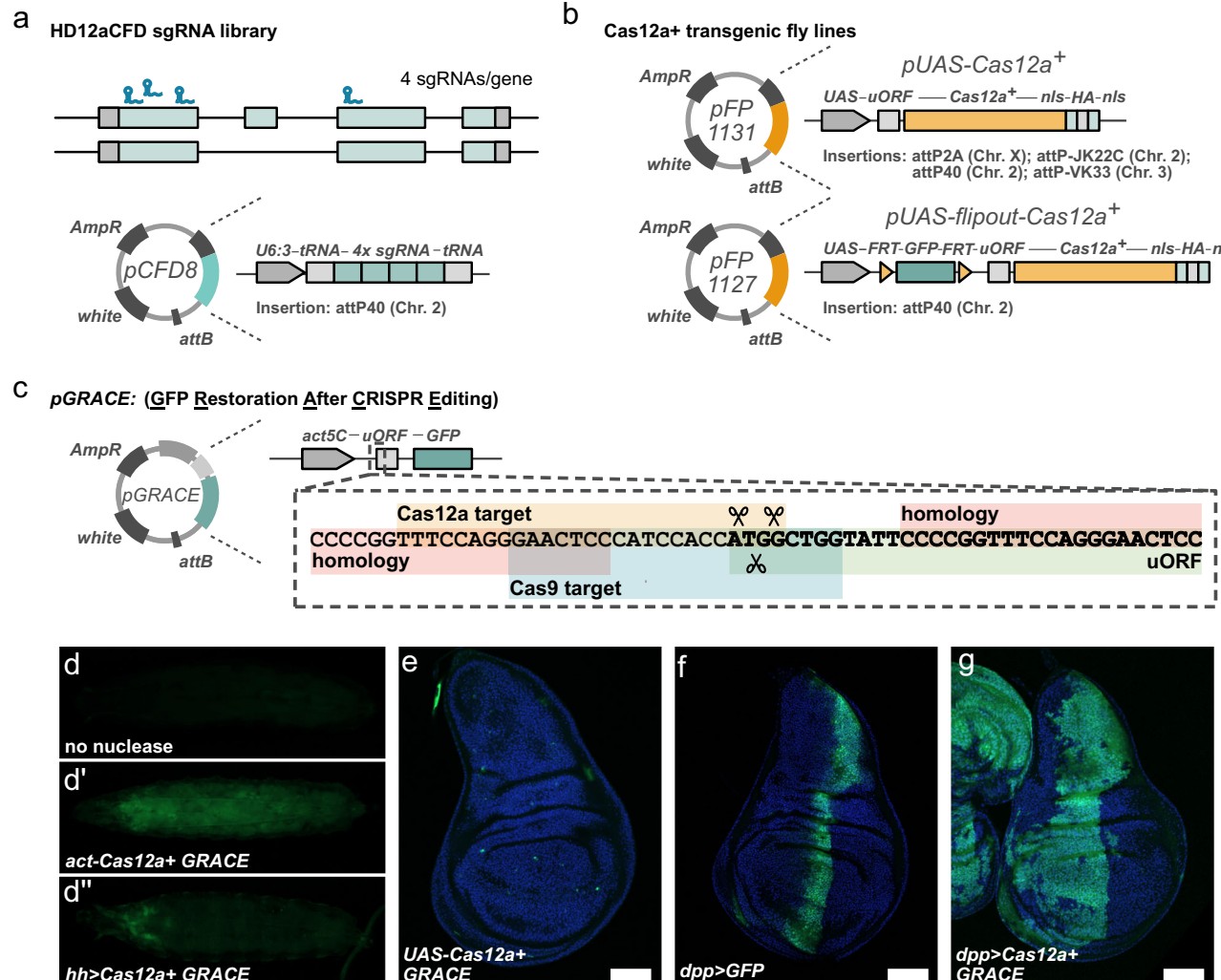

**Fig. 2 | Tools for multiplexed, conditional Cas12a⁺ gene targeting. a** HD12aCFD sgRNA vectors encode arrays of four sgRNAs targeting independent sites in common exons of the target gene. They are expressed from *pCFD8*, which contains the strong, ubiquitous *U6:3* promoter. **b** Cas12a⁺ expression vectors feature a *UAS* promoter, for Gal4-dependent expression, an upstream open reading frame (uORF), and dual nuclear localisation signals to control expression levels and subcellular localisation. Construct *pFP1127* in addition harbors a flip-out cassette, enabling control through a FLP recombinase. Cas12a⁺ and HD12aCFD fly strains are available from (https://vdrc.at). **c** *pGRACE* (pFP1128) is a reporter to visualize gene editing activity in vivo. It encodes a ubiquitous *act5C* promoter and a GFP open reading frame separated by an uORF, which suppresses GFP translation. Upon CRISPR gene editing the start codon of the uORF is removed and GFP expression is derepressed. Detailed sequence of the Cas12a/Cas9 sgRNA target regions and homology regions to bias repair is shown in the expanded view. **d–d''** GFP expression in larva without nuclease (**d**), with *act5C-Cas12a⁺ GRACE* (**d'**), and *hh-Gal4 UAS-uᴹCas12a⁺nls²ˣ GRACE* (**d''**). All genotypes also express the Cas12a⁺ GRACE-sgRNA. GRACE readily reveals ubiquitous Cas12a activity when expressed from the *act5C* promoter and tissue-specific editing when expression is driven by *hh-Gal4*. **e–g** Wing disc GFP expression patterns showing *UAS-uᴹCas12a⁺nls²ˣ GRACE* (**e**), *dpp-Gal4 UAS-GFP* (**f**), and *dpp-Gal4 UAS-uᴹCas12a⁺nls²ˣ GRACE* (**g**). In the absence of a Gal4 driver only very few cells express GFP, presumably reflecting low levels of leaky Cas12a+ expression (**e**). Acute expression of GFP from a *UAS-GFP* transgene driven by *dpp-Gal4* is visible in a domain along the anterior-posterior boundary (**f**). In contrast, GRACE reveals that mutations are induced in a much broader domain comprising most of the anterior compartment (**g**). This is likely due to broader expression of *dpp-Gal4* in early development. Scale bars: 50 μm.

## Multiplexed gene targeting does not cause widespread proximity effects in *Drosophila*

CRISPR-Cas nuclease activity can induce unintended on-target effects, including large deletions affecting multiple neighboring genes and chromosome-arm truncations that remove dozens to hundreds of genes in *cis*[19–21]. Such events can cause 'proximity effects', where gene knockouts display phenotypic similarities to unrelated genes located on the same chromosome arm[18]. We tested whether similar effects occur during Cas12a-mediated knock-out mutagenesis in *Drosophila*.

Because truncations extending from a double-strand break toward the telomere affect more genes when the break occurs closer to the centromere, proximity effects have been shown to manifest as a correlation between phenotypic impact and chromosomal position in mammalian cells[18]. To test if such a relationship is also observable in

*Drosophila*, we performed Cas12a⁺ mutagenesis in wing precursor cells using HD12aCFD arrays targeting 387 genes distributed across all chromosomes and examined the effect on wing size in relationship to target gene position (Supplementary Fig. 3). No correlation between phenotypic strength and chromosomal position was observed, suggesting that chromosome-arm-scale deletions do not occur frequently in flies under these conditions.

To increase sensitivity for detecting proximity effects, we performed a focused analysis of gene pairs located in close genomic proximity. We examined pairs in which one gene produces a well-characterized phenotype when mutated and the neighboring gene has not been functionally characterized. The uncharacterized genes were located centromeric to the characterized gene, ranging from 0 kb (overlapping on the opposite strand or intronic) to 82 kb away. If

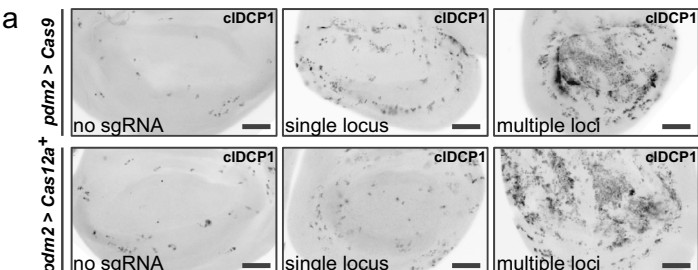

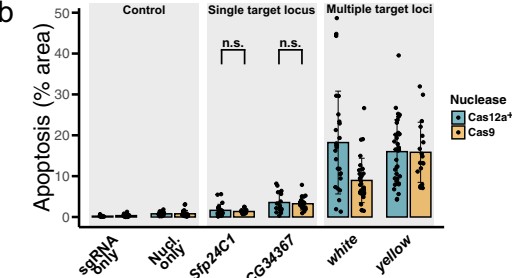

**Fig. 3 | Multiplexed CRISPR targeting shows differential toxicity based on genomic distribution of target sites. a** Examples of wing imaginal discs immunostained for cleaved death caspase 1 (clDCP1) to detect apoptotic cells. Targeting a single genomic locus (*CG34367*) results in a moderate increase in the number of apoptotic cells compared to controls without induction of DNA damage (no sgRNA). Targeting multiple chromosomal locations (*yellow*, employed as transgenic marker and hence present on all major chromosomes; see Supplementary

Fig. 2) results in a strong induction of cell death. Scale bar: 50 μm. **b** Quantification of apoptotic area (clDCP1-positive) within the nuclease expression domain (*pdm2-Gal4 UAS-Cas12a⁺*). Each point represents an individual disc; bars show mean ± SD. No significant differences (n.s.) were observed between gene targeting with Cas9 and two HD_CFD sgRNAs targeting either *Sfp24C1* (N = 13) or *CG34367* (N = 22) and Cas12a⁺ and 4 HD12aCFD sgRNAs (N = 23; N = 18)(two-sided Welch's t-test). Source data are provided as a Source Data file.

deletions extending from the target site frequently affect neighboring loci, targeting the uncharacterized genes would be expected to phenocopy knockouts of the characterized genes. We first examined 10 pairs including genes with well established phenotypes in wing morphogenesis. Targeting any of the 10 characterized genes with Cas12a⁺ and HD12aCFD sgRNA arrays produced highly penetrant phenotypes fully consistent with their known function (Fig. 4a–j). These included strong reductions in wing size in all cases, as well as more specific phenotypes, such as a failure to differentiate wing veins (*Egfr*, *tkv*, *smo*, *Mad*) or loss of margin tissue (*wg*, *Su(H)*). In contrast, targeting the uncharacterized genes in close vicinity resulted in either no effect or only mild effects on wing size and did not recapitulate any of the specific phenotypes observed when disrupting the neighboring characterized genes (Fig. 4a–j).

We complemented these findings by examining genes adjacent to *sepia*, *ebony* and *forked*, which control pigmentation or bristle morphology, respectively. In three of four cases, targeting neighboring genes produced no evidence of proximity effects (Fig. 4k–p). However, targeting *GstO3*, which is located in very close proximity to the eye pigmentation gene *sepia*, resulted in mild eye pigmentation defects with low penetrance (13%, *n* = 141, Fig. 4k, i). These phenotypes were less severe than those caused by complete *sepia* disruption, suggesting potential effects on regulatory elements rather than deletion of the *sepia* coding sequence. Notably, the closest target site in the *GstO3* sgRNA array is located just 200 bp upstream of the *sepia* transcriptional start site and therefore likely in very close proximity to such regulatory elements.

In the experiments described above, the characterized genes are recessive, meaning that both alleles would need to be affected to produce measurable phenotypes. While this applies to most genes, a small minority in the *Drosophila* genome are haploinsufficient and would therefore be more vulnerable to proximity effects. To test this more stringent scenario, we examined two HD12aCFD lines targeting genes adjacent to Minute loci, haploinsufficient genes encoding ribosomal protein subunits[38]. Disruption of a single Minute allele triggers apoptosis when mutant cells contact wild-type cells in the wing disc epithelium[39]. Directly targeting a Minute gene in wing precursor cells resulted in extensive apoptosis and lethality, with only rare escapers exhibiting severely degenerated wings (Supplementary Fig. 4). In contrast, targeting either of the two genes proximal to Minute loci caused no or only minimal apoptosis in the epithelium, no lethality, and had only minimal effects on wing size and morphology (Supplementary Fig. 4).

Taken together, these results demonstrate that multiplexed Cas12a⁺ gene targeting in *Drosophila* does not give rise to widespread

proximity effects, with effects on neighboring functional elements occurring only at low frequency or only when these elements are in very close proximity to the target site.

## Large-scale, unbiased activity screening reveals high specificity of multiplexed Cas12a⁺ gene editing

Induction of off-target mutations represents another class of potential unintended outcomes of CRISPR gene editing experiments. Given the substantial heterogeneity in off-target propensity among sgRNAs, a meaningful evaluation of gene editing specificity requires analysis across large sgRNA collections. However, suitable methods for this purpose have not been established in *Drosophila* or most other organisms. We therefore sought to develop a high-throughput assay for the unbiased detection of CRISPR nuclease activity across large sections of the genome. We focused on the visualization of loss of heterozygosity (LOH), a common repair outcome of CRISPR-induced chromosome breaks in diverse systems ranging from yeast to humans[21,40–42], and a prevalent consequence of Cas9-mediated DNA double strand breaks in *Drosophila*[33,43]. Two distinct modes of LOH have been described: copy-number neutral (CN-) LOH, which results from interhomolog recombination, and copy-number loss (CL-) LOH, which can arise through large deletions, chromosome truncations and chromothripsis (Fig. 5a). A heterozygous fluorescent reporter enables visualization and discrimination between these LOH modes: CN-LOH produces equal populations of cells that have gained or lost the reporter, whereas CL-LOH generates only reporter-loss cells (Fig. 5a, b). To determine the relative frequency of CN- versus CL-LOH in *Drosophila*, we induced double-strand breaks on a reporter-bearing chromosome arm using five distinct HD12aCFD sgRNA arrays, including two targeting in close proximity to the reporter. Quantification of fluorescence-gain and fluorescence-loss cells revealed approximately equal ratios for all sgRNA lines, indicating that CN-LOH predominates in this system (Fig. 5c, and Supplementary Fig. 5). The difference in prevalence of LOH modes between *Drosophila* and mammalian cells likely reflects known differences in chromosome biology (see Discussion) and provides an explanation for the very low frequency of proximity effects observed in the experiments described above.

LOH detection exhibits strict chromosomal specificity, as reporters respond exclusively to DNA breaks induced between their insertion site and the centromere. A reporter on chromosome arm 2 L responds only to Cas12a⁺ activity directed by sgRNAs targeting 2 L, while a 3 R reporter responds only to sgRNAs targeting on 3 R (Fig. 5d, e). Furthermore, sequencing of target sites from sgRNA arrays that induce LOH at low frequency revealed that this approach can detect low Cas12a⁺ activity, indicating suitable sensitivity for

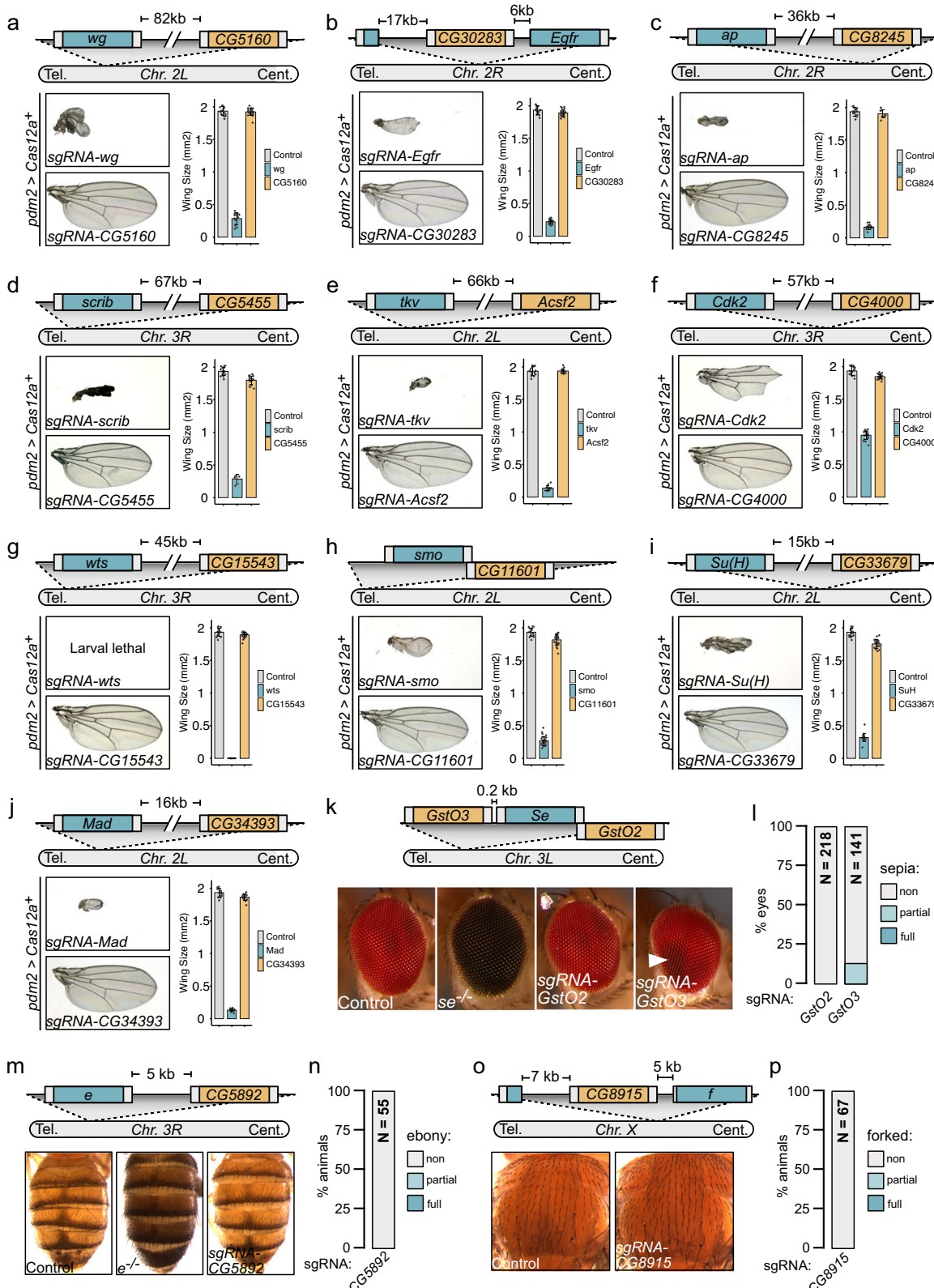

*Drosophila* research applications (Supplementary Fig. 6). Collectively, these results establish that LOH reporters enable unbiased detection of DNA break induction across entire chromosome arms and provide a sensitive readout for measuring CRISPR nuclease activity.

To enable high-throughput LOH screening, we constructed reporter genes expressing fluorophores under a tissue-specific enhancer that restricts expression to larval wing and haltere imaginal discs[44] (Fig. 5f and methods). This enables visualization of LOH in living animals with a conventional fluorescent stereoscope (Fig. 5g, h). We applied this system to screen for LOH across both arms of chromosome 2, using hundreds of HD12aCFD lines encoding over 2000 unique sgRNAs. These screens monitored CRISPR activity over 44.6 Mb, representing 33% of the *Drosophila* genome.

**Fig. 4 | Cas12a⁺-mediated multiplexed gene editing typically shows no proximity effects on neighboring genes.** Multiplexed sgRNA targeting does not disrupt adjacent genes except when these are located in very close proximity. Panels show genomic arrangements of gene pairs (schematics with distances indicated), representative phenotypes, and quantifications. Genes with unknown phenotypes are shown in orange, genes with characterized loss-of-function phenotypes in blue. Schematics indicate telomere (Tel.) and centromere (Cent.) orientation. **a–j** Targeting genes located 0-82 kb from genes controlling wing growth does not produce neighboring gene phenotypes. Bar graphs show mean wing size with individual measurements (points) and SD (error bars). Cas12a⁺ expression is restricted to the developing wing by *pdm2-Gal4*. Controls express sgRNAs targeting a gene not expressed in wings (*Sfp24C1*). Source data are provided as a Source Data file. **k, l** Extremely close genomic proximity can result in co-disruption. **k** The *GstO* genes are located in direct proximity to *sepia* (*se*). Eye images show control, *se* mutant, and Cas12a⁺-targeted phenotypes. **l** Targeting *GstO3* occasionally produces sepia eye phenotypes indicating co-disruption of the immediately adjacent *se* gene. **m–p** No detectable proximity effects for genes controlling pigmentation or bristle morphology. **m, n** Abdominal pigmentation in control, *ebony* (*e*) mutant, and animals with sgRNA targeting *CG5892* (5 kb from *e*). No *e* disruption observed (N = 55 animals). **o, p** Normal bristle morphology in both control and animals with sgRNA targeting *CG8915* (7 kb from *forked* (*f*), N = 67 animals). Experiments in (**k–p**) performed with the near-ubiquitous *act5C* promoter driving Cas12a⁺ expression.

First, we focused on sgRNA arrays designed to target genes on other chromosome arms or distal to the reporter insertion site. Because individual sgRNAs are sufficient to induce robust LOH (Supplementary Fig. 7), any array-encoded sgRNAs with off-target activity on the screened chromosome arm are expected to lead to reproducible LOH. The screen on 2 L included 1769 such sgRNAs, while the 2 R screen comprised 1605. We detected no LOH with such sgRNAs on 2 R and sporadic LOH in 16 cases in the screen on 2 L (Fig. 5i, j). The latter events could not be reproduced upon rescreening with larger sample sizes (Supplementary Fig. 8), suggesting they likely represent either stochastic background LOH or screening artefacts, although we cannot exclude the possibility of off-target activity occurring at very low frequency.

We then analysed the results from LOH screening with HD12aCFD arrays targeting genes located between the reporter insertion sites and the centromere. These were 81 lines on 2 L and 88 lines on 2 R. Only a single line failed to induce LOH (Fig. 5i, j), demonstrating highly robust on-target activity of HD12aCFD arrays.

In summary, LOH visualization enabled us to screen for Cas12a⁺ activity with over 2000 sgRNAs across 33% of the *Drosophila* genome. This constitutes, to our knowledge, the so far largest unbiased screen for nuclease activity conducted in vivo. It revealed on-target activity for over 99% of HD12aCFD arrays and uncovered potential off-target activity in less than 1% of sgRNAs, indicating excellent efficiency and specificity of gene targeting with HD12aCFD arrays.

## Multiplexed Cas12a⁺ gene targeting outperforms established Cas9-based methods

We next compared the efficiency of gene disruption with Cas12a⁺ and HD12aCFD arrays to that of current state-of-the-art Cas9-based methods. First, we performed ubiquitous or tissue-specific CRISPR mutagenesis of endogenous genes with known loss-of-function phenotypes. We examined *white* (*w*) disruption in neuronal photoreceptors, where loss of function leads to eye pigmentation defects. Strikingly, HD12aCFD-mediated gene targeting achieved near-complete *w* disruption (96 ± 5% white-eyed tissue), whereas Cas9 with a publicly available sgRNA generated highly mosaic eyes with only 50 ± 26% loss of pigmentation (Fig. 6a, b). We extended this comparison to the intestinal epithelium by inducing mutagenesis of *Notch* and *neuralized*, which are both critical regulators of intestinal stem cell proliferation. HD12aCFD targeting of either gene resulted in significantly greater expansion of mitotic cells compared to a dual-sgRNA Cas9 system (Fig. 6c, d). Similarly, targeting *smoothened* (*smo*) specifically in the pouch domain of wing imaginal disc epithelia resulted in near complete loss of Smo protein with a HD12aCFD line, while cells expressing two sgRNAs and Cas9 retained a significant number of Smo positive cells (Fig. 6e, f). Collectively, these data indicate that gene targeting with Cas12a⁺ and multiplexed sgRNAs achieves superior gene disruption efficiency across diverse tissues.

To systematically benchmark HD12aCFD performance, we developed a quantitative assay using adult wing size as a scalable phenotypic readout. We targeted genes specifically in wing precursor cells using *pdm2-Gal4* and employed automated image analysis with a custom-trained segmentation algorithm to measure wing size across multiple individuals per genotype (Supplementary Fig. 9). Applying this system to over 300 target genes, HD12aCFD-mediated targeting of known growth regulators consistently produced size reductions, while targeting of control genes left wing morphology intact, confirming both efficiency and specificity (Supplementary Fig. 9).

We next compared HD12aCFD arrays against existing CRISPR resources for gene disruption in *Drosophila* through parallel screening. We evaluated two major collections: sgRNA lines from the Bloomington Drosophila Stock Center (BDSC) and the HD_CFD library from the Vienna Drosophila Resource Center (VDRC)[10–12]. These collections encompass diverse design strategies, including different promoter choices (U6:3 or UAS) and sgRNA numbers (one or two), representing the range of editing approaches currently in use. For direct comparison, we obtained all available BDSC and VDRC lines targeting genes also targeted by HD12aCFD lines and performed side-by-side wing-specific mutagenesis.

HD12aCFD arrays significantly outperformed both existing resources when tested against 154 BDSC and 103 VDRC lines (p < 0.001, Wilcoxon signed-rank test; Fig. 6g, h). As expected, many genes showed no wing phenotype with either system, indicating that these genes are not required for wing development. Among genes that produced phenotypes and thus provide informative comparisons for assessing knockout efficiency, effects ranged from comparable between systems to substantially stronger with HD12aCFD arrays. Importantly, this included several established wing growth regulators such as *wg*, *ap*, *tkv*, *Su(H)*, *Raf*, and *Cdk1*, which produced minimal to moderate phenotypes with Cas9-based systems but resulted in strong size reductions with HD12aCFD arrays. This demonstrates that these outliers represent functional false negatives in conventional Cas9-based screening approaches and establishes HD12aCFD technology as a substantial advance for robust gene disruption.

Finally, we tested whether a recently developed Cas9 sgRNA expression vector featuring an optimized scaffold and efficient tRNA-based processing[45] could close the performance gap between Cas9 and HD12aCFD systems. Since this improved design has not yet been incorporated into the large-scale resources evaluated above, we tested it separately on three morphogenetic regulators where HD12aCFD arrays had faithfully recapitulated expected loss-of-function phenotypes, while HD_CFD lines produced weaker effects. We cloned the same sgRNAs used in the HD_CFD lines into the improved expression vector and inserted them at the same genomic landing site, eliminating sgRNA selection and expression as a confounding variable. While the improved vector modestly enhanced phenotypic penetrance for two of three genes compared to HD_CFD lines (Supplementary Fig. 10), it still fell substantially short of HD12aCFD performance across all targets. These controlled comparisons demonstrate that HD12aCFD technology surpasses even the latest optimizations in Cas9-based systems.

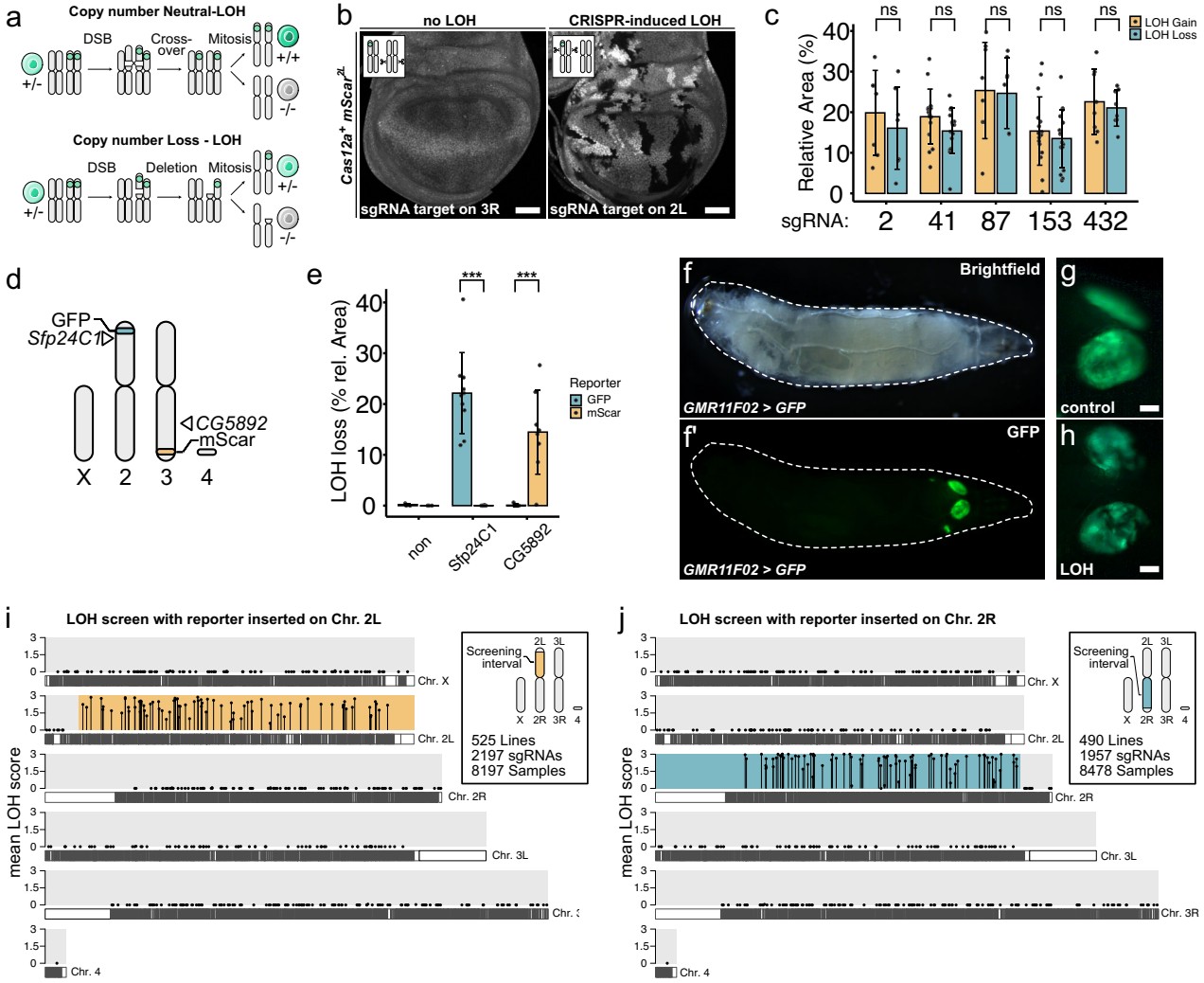

**Fig. 5 | Visualization of CRISPR-induced loss of heterozygosity (LOH) enables unbiased detection of Cas12a+ activity and reveals high specificity.** LOH, a common outcome of CRISPR-induced chromosome breaks, enables unbiased detection of nuclease activity across large genomic regions. **a** Schematic of LOH mechanisms. Copy number neutral (CN-) LOH occurs through mitotic recombination or reciprocal end joining, producing homozygous daughter cells (+/+ or -/-) from heterozygous parents (+/-). Copy number loss (CL-) LOH results from large deletions or chromothripsis. A heterozygous fluorescent reporter enables tracking of LOH events. CN-LOH also generates cells carrying two copies of the reporter, whereas CL-LOH exclusively leads to reporter loss. **b** CRISPR-induced LOH in larval wing imaginal discs expressing Cas12a +, sgRNA arrays, and a heterozygous mScarlet (mScar) reporter on 2 L. Targeting 3 R does not cause reporter LOH (left), while sgRNAs targeting between the mScar transgene and centromere induce frequent LOH (right). Scale bar: 50 μm. **c** CN-LOH predominates in wing discs. Five sgRNA arrays mediate equal loss/gain ratios. Points = individual discs aks, bars = mean, error bars = SD. No significant differences were observed (two-sided paired t-

test, FDR correction). For genomic distances and representative images see Supplementary Fig. 4. **d** Dual-reporter schematic: GFP (2 L) and mScar (3 R) enable simultaneous LOH detection on different chromosome arms. **e** Chromosome arm-specific LOH. sgRNA arrays targeting Sfp24C1 (2 L) or CG5892 (3 R) induce LOH exclusively on their respective arms. ***P < 0.001 (two-sided paired t-test). **f**–**h** High-throughput LOH screening system. (f) GMR11F02 drives selective GFP expression in wing and haltere disc pouches. (**g**) Control showing continuous GFP expression. (**h**) LOH detected as GFP-negative patches in living larvae. Scale bars: 50 μm. **i**, **j** High-throughput screening on 2 L and 2 R reveals high on-target specificity. Mean LOH scores (10-20 discs discs) plotted at genomic locations. Screening intervals (reporter to centromere) indicated in orange (2 L) or blue (2 R). 8197 discs screened with 525 lines (2197 sgRNAs) for 2 L; 8478 samples with 490 lines (1957 sgRNAs) for 2 R. 168/169 sgRNA arrays targeting genes within screening intervals produced reproducible LOH. No arrays targeting outside screening intervals caused reproducible LOH (sporadic LOH in 16 lines; see Supplementary Fig. 8). Source data are provided as a Source Data file.

## Increased gene disruption efficiency reveals a somatic function for *trade embargo*

The mutagenesis screens described above also included genes with no previously documented role in wing morphogenesis. One of them was *trade embargo* (*trem*), which encodes a C2H2 zinc-finger protein. Mutagenesis of *trem* with a HD12aCFD sgRNA array resulted in a 24.5 ± 3.8% reduction in wing size, while gene targeting with a HD_CFD sgRNA pair produced no such phenotype (4.3 + − 4.9% reduction in wing size, Fig. 7a, b). Additionally, we observed a highly penetrant defect in wing vein differentiation in animals

expressing Cas12a⁺ and *HD12aCFD-trem*, characterized by absence of the posterior cross vein and defects in the L2, L4, and L5 veins (Fig. 7a″, c, d). No such vein defects were detected when mutagenesis was performed with Cas9 and the *HD_CFD-trem* sgRNA construct (Fig. 7a′, c, d).

The wing phenotypes observed exclusively after multiplexed Cas12a⁺ targeting of *trem* could result from either more efficient gene disruption or potential artifacts associated with the *HD12aCFD-trem* line. To distinguish between these possibilities, we generated two additional Cas9 sgRNA pairs. These new constructs

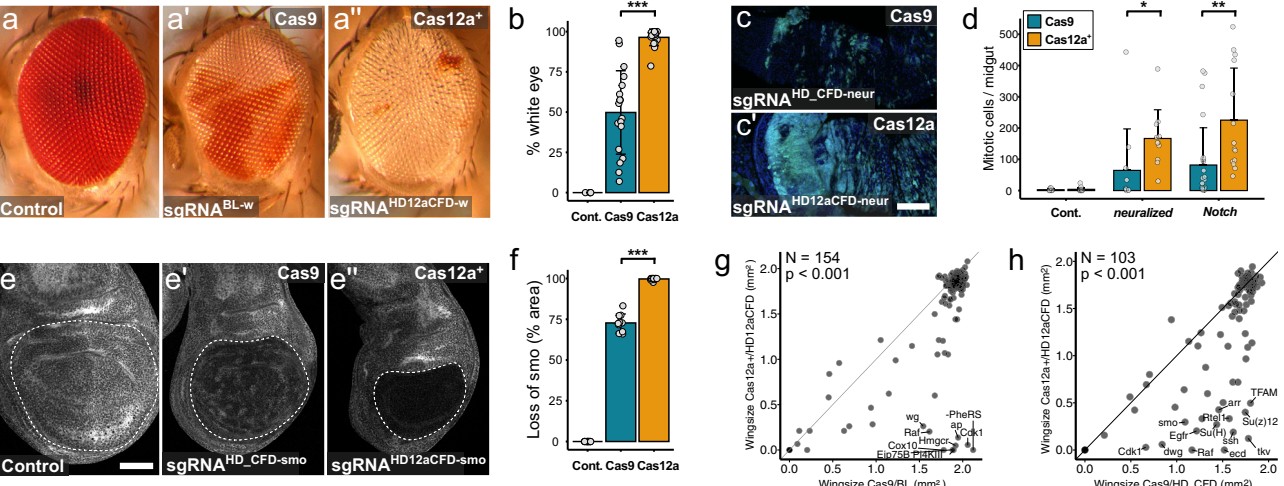

**Fig. 6 | Cas12a+ and HD12aCFD arrays enhance gene knock-out efficiency compared to current Cas9-based systems. a, a″** The HD12aCFD system improves disruption of *white* (*w*) in the eye. Representative images of eyes from flies with no sgRNA (control) or *act-Cas9* and a single sgRNA or *act-Cas12a+* and a HD12aCFD array. All genotypes contain three functional copies of *w* due to transgene markers. **b** Quantification of white eye area. Individual data points show white area per eye, with mean (columns) and standard deviation (error bars). **p < 0.001, two-sided independent two-sample t-test. **c, c′** Posterior *Drosophila* midgut showing stem and progenitor cells (ISCs/EBs, green) and DNA (blue). Targeting *neuralized* (*neur*) with Cas12a + /HD12aCFD using an inducible esg^TS system increases ISC/EB numbers. Scale bar: 50 µm. **d** Quantification of mitotic cells (phospho-Histone3 positive) in the midgut. Loss of *neur* or *Notch* function leads to an increase in cell proliferation, with significantly stronger effects using Cas12+ and HD12aCFD sgRNA arrays. Points: mitotic cells per midgut; columns: mean per condition; error bars: standard

deviation; **p < 0.01, *p < 0.05, two-sided independent two-sample t-test. **e, e″** Immunofluorescent staining of Smo in wing imaginal discs. Conditional mutagenesis in the pouch region (dotted line) with *pdm2-Gal4 UAS-Cas12a+* and HD12aCFD sgRNAs results in more effective gene knock-out. Note correlation between knock-out efficiency and pouch size. Scale bar: 50 µm. **f** Quantification of Smo protein loss. Data points show affected area per disc, with mean (columns) and standard deviation (error bars). ***p < 0.001, two-sided independent two-sample t-test. **g, h** Comparative analysis of wing size phenotypes resulting from tissue-specific gene targeting with different CRISPR systems. Cas12a+ with HD12aCFD sgRNA arrays results in stronger phenotypes than Cas9 editing with BL sgRNA (**g**) or HD_CFD sgRNA (**h**). Zero values indicate lethal phenotypes. Systems show significantly different effects on tissue size (Wilcoxon signed-rank test, V = 10045 in (**g**) and 3915 in (**h**), *n* = 154 in (**g**) and 103 in (**h**), *p* < 0.001, two-tailed). Source data are provided as a Source Data file.

produced wing size and vein defects when combined with tissue-specific Cas9 expression (Fig. 7b–f). Although these phenotypes were less severe than those observed with *HD12aCFD-trem*, they recapitulated all aspects of the wing phenotype: size reduction, loss of the posterior cross vein, and defects in the lateral longitudinal veins. These results demonstrate that the observed phenotypes arise from *trem* disruption and are not specific to any particular sgRNA line.

A role for *trem* in wing morphogenesis was surprising, as it was not reported in a previous study focusing on the function of *trem* in *Drosophila*[46]. This study analysed a transposon insertion allele, *trem^{f05891}*, which was classified as loss-of-function based on undetectable Trem protein levels using a specific antibody. However, the *trem* coding sequence remains intact in *trem^{f05891}*, raising the possibility that instead it could be a strong hypomorphic allele (Fig. 7g). We therefore generated new *trem* alleles through CRISPR gene targeting in the germline. We recovered two indel alleles in the *trem* coding sequence that disrupt the protein reading frame. After backcrossing both alleles for multiple generations to eliminate potential secondary mutations, we combined each allele with a balancer chromosome carrying a dominant marker and tested their homozygous viability. In contrast to *trem^{f05891}*, which is homozygous viable without somatic phenotypes, both *trem* CRISPR alleles are homozygous lethal, with animals lacking balancer chromosomes dying at mid-larval stage (Fig. 7h).

Together, these findings establish *trem* as an essential gene in *Drosophila* with critical roles in growth and patterning of wing appendages. These functions remained undetected with existing germline alleles and an established Cas9 sgRNA line, as these tools did not achieve sufficient levels of gene inactivation to reveal the full spectrum of *trem* activities.

## Discussion

In this study, we developed an improved CRISPR gene ablation system. Our approach combines Cas12a+ with four arrayed sgRNAs targeting independent sites in each gene to overcome a general limitation of gene knockouts with CRISPR nucleases in multicellular systems - the generation of genetic mosaics that include cells with functional gene copies. Direct comparisons with current Cas9-based approaches demonstrate that this system not only increases knockout penetrance across diverse tissues, but also reveals phenotypes previously missed by established approaches. Importantly, systematic evaluation of potential limitations showed that the method maintains high specificity despite increased sgRNA multiplexing and is well tolerated in vivo, making it a robust platform for functional studies.

The superior performance of Cas12a+ with quadruple sgRNA arrays stems from both redundant and synergistic effects of multiplexed targeting. Redundancy of sgRNAs ensures robust editing, with activity observed in >99% of sgRNA arrays in our LOH screens, a marked improvement over established single and double Cas9 sgRNA libraries that typically include 10-20% inactive lines[10–12]. Additionally, multiplexed sgRNAs can act synergistically by creating deletions between target sites. Induction of such deletions requires near-simultaneous DNA cutting, which increases in frequency when more sgRNAs target the same locus.

Intrinsic differences in cleavage activity and kinetics between the Cas9 and Cas12a variants used here may contribute to the performance differences observed in our experiments[47]. However, the current study was not designed to directly compare the catalytic capabilities of these two enzymes. Such a comparison would require arrays with identical sgRNA numbers and closely matched target sites, the latter being substantially complicated by differing PAM requirements. While construction of quadruple Cas9 sgRNA arrays has been described[4,6],

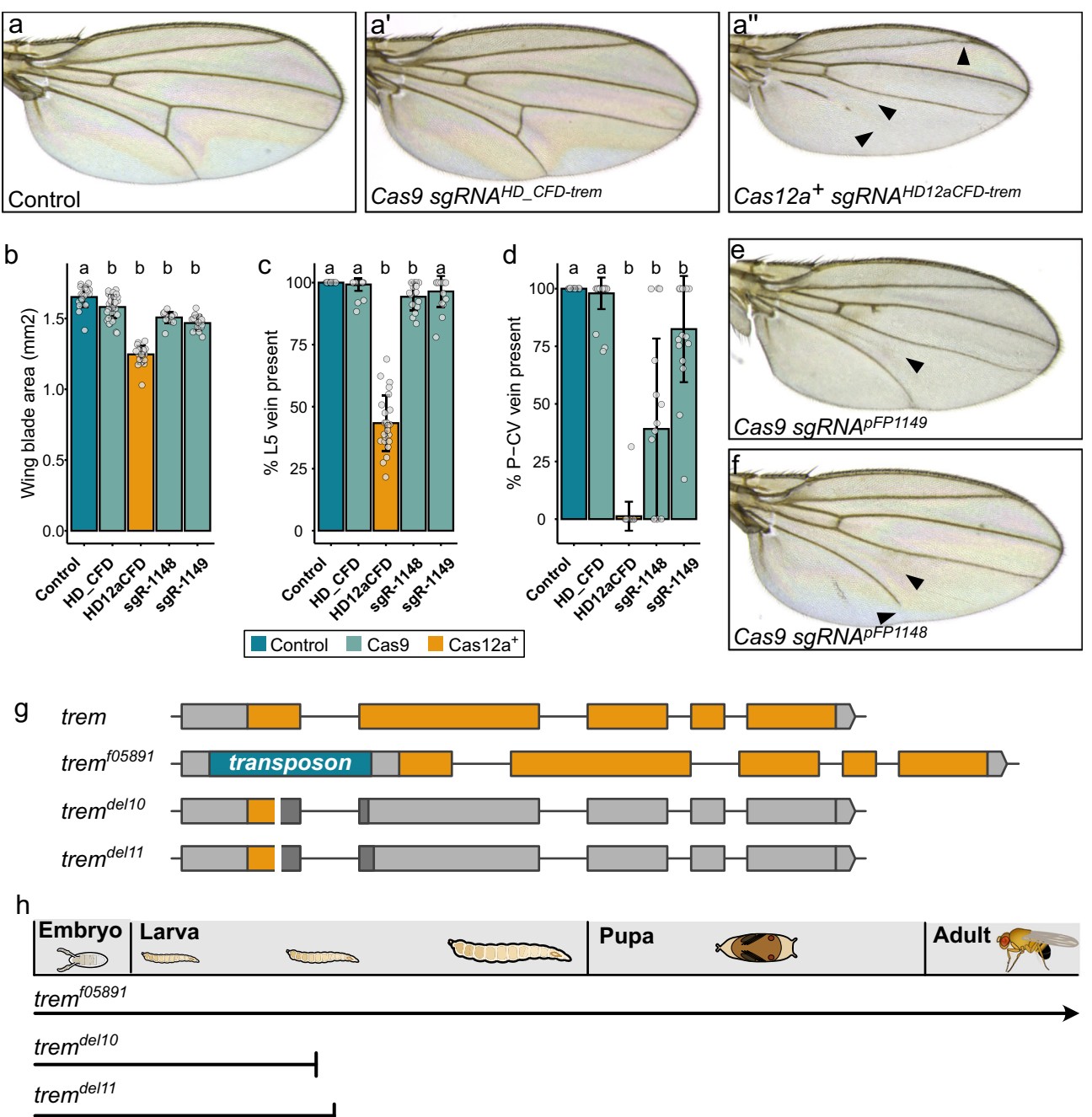

**Fig. 7 | Multiplexed Cas12a⁺ targeting reveals essential functions of *trem* in wing development. a**, **a"** Wing phenotypes following *trem* targeting. Control (**a**), Cas9 with two sgRNAs (**a'**), and Cas12a⁺ with 4x sgRNA array (**a"**). Cas12a⁺ targeting causes penetrant vein defects (arrowheads) affecting L5 and P-CV2 veins, with variable L2 and L4 disruption. **b** Wing blade area quantification. Cas12a⁺ targeting causes stronger size reduction than Cas9. **c**, **d** Quantification of L5 vein and P-CV2 length. Multiplexed Cas12a⁺ targeting shows fully penetrant vein defects, while targeting with Cas9 and sgRNA pairs results in weaker phenotypes. In **b–d** data are presented as mean ± SD with individual data points overlaid. Statistical comparisons to control were performed using one-way ANOVA with Dunnett's post-hoc test. Conditions labeled 'a' are not significantly different from control; conditions labeled 'b' are significantly different from control ($p < 0.05$). Source data are provided as a Source Data file. **e**, **f** Independent validation using two Cas9 sgRNA²ˣ lines (*pFP1149*, *pFP1148*), showing similar but weaker vein phenotypes (arrowheads). **g** *trem* gene structure and mutant alleles. *trem^f05891*: 5′UTR transposon insertion preserving coding sequence; *trem^del10* and *trem^del11*: CRISPR-induced frameshift mutations. **h** Developmental progression of *trem* mutants. *trem^f05891* homozygotes are viable[46], while frameshift mutations cause mid-larval lethality.

scaling this approach remains cost- and labor-intensive. A recently described quadruple Cas9 sgRNA library for the human genome, for example, required assembly from over 100,000 individual PCR reactions using individually synthesized oligonucleotides[6]. Given such practical constraints and the fact that equivalent Cas12a libraries are substantially easier to construct, it appears unlikely that large Cas9 sgRNA libraries with more than two sgRNAs per gene will be generated for use in *Drosophila*. Therefore, we benchmarked the Cas12a library described here against the Cas9-based designs currently available to researchers using the fly as a model organism and found it provides a major improvement in gene knockout efficiency.

Possible adverse effects of sgRNA multiplexing include toxicity from excessive DNA damage. Studies in mammalian cells have shown that cells with multiple target sites activate DNA damage checkpoints

and are lost in pooled CRISPR screens[48,49]. However, in these cases, the gene copies are usually dispersed in the genome. We find that increasing sgRNA number from two to four at a single locus does not detectably increase cell death. In contrast, targeting gene copies on multiple chromosomes, even with fewer total cut sites, significantly increases apoptosis and causes severe adult phenotypes. This suggests that the spatial distribution of cut sites, rather than their total number, is a major determinant of toxicity, possibly through the formation of large deletions and chromosomal translocations when targeting multiple loci.

Proximity effects represent another potential detrimental consequence of CRISPR nuclease-induced chromosome breaks. These effects can arise through large deletions emanating from cut sites[20], truncations of entire chromosome arms[18], or chromothripsis[19], leading to loss of neighboring genes located *in cis* to the target gene. Recent work in mammalian cells has demonstrated that such events create systematic proximity bias in large-scale CRISPR screens, where genes in physical proximity are more likely to share similar phenotypes. However, we find no evidence for widespread proximity effects in *Drosophila*. Loss-of-function phenotypes do not correlate with chromosomal position of target genes, and detailed analysis of gene pairs in close proximity reveals no frequent disruption of neighboring loci. These differences likely reflect fundamental distinctions in chromosome biology between dipterans and mammals. In *Drosophila*, homologous chromosomes pair in both somatic and germ cells, whereas in mammals, homolog pairing is restricted to meiosis[50,51]. Consequently, in flies the homologous chromosome is in close physical proximity to a CRISPR-induced DNA double-strand break and is more likely to serve as repair template or to engage in reciprocal end joining. Supporting this hypothesis, we find that interhomolog recombination is highly prevalent following Cas12a gene targeting and represents the dominant mode of loss of heterozygosity in flies, consistent with recent studies on Cas9-induced chromosome breaks[33,43].

While our results indicate that proximity effects are not a widespread problem in CRISPR mutagenesis experiments in *Drosophila*, they may occasionally confound results when functional elements lie in close proximity to target sites or are embedded within target genes. For instance, regulatory elements such as enhancers can reside within introns of other genes and may be disrupted by deletions induced at target loci. Additionally, interhomolog recombination can reveal recessive phenotypes of pre-existing mutations and complicate experiments involving heterozygous transgenes. Nevertheless, experimentally induced mitotic recombination has been extensively used in *Drosophila* research[52,53] and has proven compatible with meaningful biological discoveries across hundreds of studies.

Multiplexing several sgRNAs could theoretically increase off-target effects. Because off-target activity varies widely between individual sgRNAs, meaningful characterization requires analysis of many guides. However, high-throughput assays to detect CRISPR activity at unknown genomic loci have been lacking. Our method of visualizing LOH in living animals fills this technical gap. Since LOH is a widespread repair outcome of DSB repair in diploid organisms[21,40–42], this assay could be broadly applicable to other systems. Screening for nuclease activity across 33% of the *Drosophila* genome in combination with over 2000 sgRNAs failed to detect any reproducible cases of off-target activity. While low-frequency off-target mutations remain a possibility and need to be controlled for by appropriate experimental design, this confirms that the high specificity of Cas12a observed in vitro[54] is maintained during in vivo multiplexed editing.

*Drosophila* was the first organism in which targeted gene editing was achieved[55] and has proven to be an excellent model for genome engineering approaches that later proved valuable in other animals[4,56]. This broad transferability stems from the fact that Cas proteins function effectively across different organisms, and many general principles of DSB repair are conserved during evolution. Beyond establishing a new generation of CRISPR libraries for gene disruption in *Drosophila*, the improved efficiency and high specificity of multiplexed Cas12a gene targeting has therefore implications for diverse applications. For example, current gene drive systems, which hold promise for controlling vector-borne diseases, are limited by the emergence of resistance alleles. Recent studies have demonstrated that sgRNA multiplexing can substantially delay resistance[57,58] suggesting that our system could accelerate the development of more robust genetic tools for disease control. Ultimately, this work not only provides a blueprint for developing enhanced gene knockout systems across multicellular organisms but also contributes to a broader toolkit for addressing both fundamental questions in biological research and pressing societal needs.

## Methods

### Plasmids
**Plasmid construction.** Plasmid cloning was performed using PCR amplicons generated with Q5 PCR polymerase (New England Biolabs). DNA fragments were assembled by either In-Fusion cloning (Takara) or by GoldenGate cloning using BbsI-HF and T4 DNA Ligase (both New England Biolab) using standard procedures or by extension overlap PCR (see below). PCR conditions followed manufacturer's recommendations. Individual oligonucleotides for cloning were ordered from Integrated DNA Technologies in standard, desalted format. Newly cloned inserts were confirmed by Sanger sequencing. Sequences of plasmids and primers can be found in Supplementary Data 2. Plasmids are available from the European Plasmid Repository (https://www.plasmids.eu) and Addgene (https://www.addgene.org) with accession IDs provided below for each plasmid.

**pSTAR(1–6x).** The reporter plasmids for multiplexed cutting (sgRNA target site array (STAR)) were generated by cloning a fragment of the *lacZ* coding sequence into a plasmid backbone containing *AmpR* and *mini-white* selection genes, as well as *attB* and *ori*, that was derived by BamHI digestion of plasmid *pCFD6*[4]. The *lacZ* fragment lacks both a promoter and start codon and is presumed to be transcriptionally inactive. Afterwards, the sequence TTTCCAGGGAACTCCCATCCACCATGGCTGGTATTCC, which can be targeted by a Cas12a sgRNA (CAGGGAACTCCCATCCACCATGG) or Cas9 sgRNA (GAACTCCCATCCACCATGGC), was introduced by overlap extension PCR. Two PCR amplicons were generated with an overlapping sequence containing the sgRNA target site (see Supplementary Data 2 for primer sequences). Equimolar amounts of the two PCR amplicons were mixed and subjected to PCR for 15 cycles. Then, primers annealing to the end of the fused PCR amplicon were added and full length products amplified for another 20 PCR cycles. PCR amplicons were separated on a 1% agarose gel, full length amplicons were excised, purified and cloned into BamHI digested pCFD6 by In-Fusion cloning.

**act5C-mScarlet.** A double-stranded DNA fragment encoding the *act5C* promoter, *mScarlet* coding sequence and *act5C* 3′UTR was synthesized from Integrated DNA Technologies. It was assembled with a plasmid backbone fragment containing *AmpR* and *mini-white* selection genes, as well as *attB* and *ori*, which was obtained by BamHI digestion of plasmid *pCFD6*[4] by In-Fusion cloning. The plasmid is available from the European Plasmid Repository (EPR#897) and Addgene (Plasmid 230906).

**GMR11F02-Gal4 UAS-GFP.** A fragment encoding the *GMR11F02* regulatory element and Gal4 coding sequence was PCR amplified with primers *GMR11F02G4fwd* and *GMR11F02G4rev* from plasmid *pGMR11F01-Gal4* (a gift from Ryan Loker and Richard Mann, Columbia University, New York). The fragment was then inserted into *pUAST-EGFP* digested with XhoI by In-Fusion cloning. The plasmid is available

from the European Plasmid Repository (EPR#898) and Addgene (Plasmid 230907).

**UAST-u^MCas12a⁺nls^1x**. Plasmid *UAST-u^MCas12a⁺nls^1x* was derived from *UAS-LbCas12a*[25]. An upstream open reading frame was PCR amplified from *UAS-u^MCas9*[10] with primers *uMCas12afwd* and *uMCas12arev* and gel purified and cloned into *UAS-LbCas12a* digested with EcoRI by In-Fusion cloning.

**UAST-FRT-GFP-FRT-u^MCas12a⁺nls^1x**. Plasmid *UAST-FRT-GFP-FRT-u^MCas12a⁺nls^1x* was derived from *UAST-u^MCas12a⁺nls^1x*. The plasmid was digested with EcoRI-HF and HindIII and the sequence encoding the FRT-GFP-FRT cassette was synthesized by Integrated DNA Technologies and inserted by In-Fusion cloning. The plasmid is available from the European Plasmid Repository (EPR#899) and Addgene (Plasmid 230908).

**UAST-u^MCas12a⁺nls^2x**. Plasmid *UAST-u^MCas12a⁺nls^2x* was derived from *UAST-u^MCas12a⁺nls^1x*. The sequence encoding an additional C-terminal nucleoplasmin nuclear localization signal was ordered as double-stranded DNA from Integrated DNA Technologies and inserted into *UAST-u^MCas12a⁺nls^1x* digested with NdeI and HpaI by In-Fusion cloning. The plasmid is available from the European Plasmid Repository (EPR#900) and Addgene (Plasmid 230909).

**pGRACE**. Plasmid *pGRACE* was derived from pAct5c-LbCas12a⁺[29]. The backbone was linearized with EcoRI and XhoI. The sequence encoding the uORF with binding sites for Cas12a and Cas9 sgRNA flanked by microhomology and the fluorescent protein was synthesized by Integrated DNA Technologies and inserted by In-Fusion cloning. The plasmid is available from the European Plasmid Repository (EPR#901) and Addgene (Plasmid 230910).

**Cloning of sgRNA plasmids**. Individual sgRNA plasmids were constructed as previously described[29]. Briefly, Cas12a sgRNA arrays were cloned into the BbsI site of the expression vector *pCFD8* (European Plasmid Repository EPR#32; Addgene #140619). For individual sgRNAs the spacer sequence was ordered as two individual oligonucleotides encoding the top and bottom DNA strand, including appropriate overhangs, phosphorylated, annealed and ligated into *pCFD8*. To clone individual plasmids encoding sgRNA arrays, the array sequence was split into several oligos, with the split being located in the middle of the unique spacer sequences. Oligos were phosphorylated, assembled with 'bridge' oligos, which are reverse complementary to the junctions of two oligos, and ligated by T4 DNA ligase (New England Biolabs). Subsequently, ligated oligos encoding the full sgRNA array were amplified by PCR and cloned into the BbsI site of pCFD8 by GoldenGate cloning. We have published a detailed step-by-step cloning protocol in the supplementary material of reference[29]. The spacer sequences of sgRNAs are listed in Supplementary Data 1.

**sgRNA library design**
HD12aCFD sgRNAs have been designed using the CRISPR Library Designer (v. 1.5)[32]. Guides were designed against the *D. melanogaster* genome release BDGP6.22, allowing only guides in coding sequences with less than 10 predicted off-targets of an edit distance larger than 2. For aligning crRNA target sites back to the reference genome Bowtie was used in very-sensitive mode and the first 3 PAM distal bases were ignored. The PAM requirement was set to a 5' adjacent TTTN. As basis for design served a list of ENSEMBL identifiers covering all protein coding genes in BDGP6.22. Raw, designed crRNA targets were then further processed in a custom R pipeline to yield optimized crRNA target pairs for combinatorial mutagenesis as follows.

The CLD output files were filtered according to the design criteria. In the following steps, relevant columns from the large CLD output table were selected. Everything irrelevant to the list of target genes was removed, and the 'Number of Hits' column was adjusted to represent off-target count by subtracting one count. After that, all sequences containing BbsI (GAAGAC) and BsaI (GGTCTC) restriction sites, non-unique crRNA targets, and crRNA targets adjacent to a TTTT PAM were removed. Additional selection criteria for the remaining pool of sgRNA designs were annotated as follows: the number of targeted protein-coding transcripts, whether the sgRNAs target the first half of the target gene, and the GC content of sgRNA sequences. Only those sgRNAs that hit at least max(#transcripts)-1 transcripts were kept, and the average gene expression for each transcript hit by a putative crRNA was annotated. Designs that unintentionally also hit a coding sequence of a different gene than they were designed for were removed. Further, small genes that are contained in overarching giant genes, like *kirre*, or *Raf*, were excluded. Each sgRNA was annotated on how many CDS it targets and if it targets in the first half of the target gene. To check the pool for sanity, the sgRNAs were written in GFF3 format. This file could then be loaded into an arbitrary genome browser. Next, 180 random pairs of sgRNAs were drawn for each gene and scored by optimizing pair distance, microhomology score[59], exon expression, isoform coverage, off-target counts, GC content, and exon position relative to the transcript. The two top-scored crRNA pairs were chosen for further quality controls by manual and automatic inspection.

**Cloning of the HD12aCFD sgRNA library**
Cas12a sgRNA arrays were cloned into the BbsI site of the expression vector *pCFD8* (European Plasmid Repository EPR#32; Addgene #140619), which has been previously described[29]. Oligonucleotides encoding four sgRNAs, as well as primer binding and BbsI restriction sites at either end were synthesized by Twist Bioscience. The oligonucleotide design can be found in Supplementary Data 2. Oligos were resuspended in dH₂O and amplified by PCR using primers *Cas12arrayAMPfwd2* and *Cas12arrayAMPrev2*. PCR products were separated on a 1% agarose gel, excised, purified and cloned into *pCFD8* by GoldenGate cloning using standard conditions. DNA was transformed into chemically competent E. coli (Stellar Silver competent cells, Takara) and plated on LB agar plates containing Carbicilin. After overnight incubation at 37 C, ten individual colonies were picked for validation purposes and all other colonies were pooled. DNA was extracted using QIAprep Spin Miniprep Kit (Qiagen) following manufacturer's instructions. DNA from individual colonies was validated by Sanger sequencing. When at least 9 of the 10 tested colonies contained unique, correct sgRNA plasmids we proceeded with injection of the DNA from the pooled colonies into *nos-PhiC31; attP40* transgenic *Drosophila* embryos.

**Visualization of CRISPR-edited PCR amplicons**
To visualize diversification of the target locus by CRISPR editing, reflecting the heterogeneity of the edited sequences, genomic DNA was extracted from individual flies as previously described[29]. Flies were collected in PCR tubes containing 30 μL squishing buffer (10 mM Tris-HCl pH8, 1 mM EDTA, 25 mM NaCl, 200 μg/mL Proteinase K). Flies were disrupted in a Bead Ruptor (Biovendis) for 20 s at 20 Hz. Samples were then incubated for 30 min at 37 °C, followed by heat inactivation for 3 min at 95 °C. Typically, 2 μL of supernatant were used in 30-μL PCR reactions. The target locus was amplified by PCR using locus-specific primers (Supplementary Data 2) with Q5 High-Fidelity DNA Polymerase (New England Biolabs) and no more than 30 PCR cycles under standard conditions. Amplicons were separated on 1% agarose gels. Note that amplicon pools containing diverse CRISPR edits typically form heteroduplexes during PCR, which results in the formation of a smear towards higher- and lower molecular weight products on the gel.

## Identification of CRISPR alleles by Sanger sequencing

To identify individual CRISPR-induced alleles mutagenesis was performed in the germline and genomic DNA was extracted from individual offspring as described above and the target locus was amplified by PCR using locus-specific primers (Supplementary Data 2) that annealed typically 300−500 bp 5' and 3' of the sgRNA target site. PCR amplicons were purified using paramagnetic beads[60] and sent for Sanger sequencing (Eurofins Genomics). Reads were only excluded from analysis if sequencing quality was insufficient for unambiguous base calling.

## Fly stocks and husbandry

The list of target genes and corresponding sgRNAs for which we generated HD12aCFD transgenic lines is reported in Supplementary Data 1. All other transgenic *Drosophila* strains used or generated in this study are listed in Supplementary Data 3. Fly stocks are made publicly available through the non-profit Vienna Drosophila Resource Center (https://vdrc.at). *Drosophila* were cultured on standard cornmeal food media in an incubator set to 27 °C with a 12 h light−12 h dark cycle and 60% relative humidity. Adult flies were transferred to fresh food vials every 3−4 days.

## Transgenesis

Transgenesis was performed with the PhiC31/attP/attB system and plasmids were inserted in landing sites on the second or third chromosome as indicated for each line. Microinjection of plasmids into *Drosophila* embryos was carried out using standard procedures. Transgenesis of crRNA plasmids was typically performed by a pooled injection protocol, as previously described[29]. Briefly, individual plasmids were pooled at equimolar ratio and DNA concentration was adjusted to 250 ng/μL in dH$_2$O. Plasmid pools were microinjected into blastoderm embryos, raised to adulthood and individual flies crossed to P{ry[+t7.2] = hsFLP}1, y[1] w[1118]; Sp/CyO-GFP or P{ry[+t7.2] = hsFLP}1, y[1] w[1118];; Sb/TM6B. Transgenic offspring were identified by eye color and individual transgenic flies from pooled plasmid injections were genotyped as previously described[24]. Briefly, genomic DNA was extracted from individual flies as described above and the sgRNA locus was amplified by PCR using primers (pCFD8genofwd5: TTAACGTCGGGGCTTTGAGT,pCFD8genorev6: CGACACTAGTGGATCCGTTGT). Flies were crossed to balancer flies, and stable transgenic stocks were generated.

## Immunofluorescence staining

*Drosophila* larvae were dissected in ice-cold PBS and fixed in 4% paraformaldehyde for 25 minutes at room temperature. Tissue was washed with PBT (PBS with 0.3% Triton-X100) and blocked with 1% heat-inactivated normal goat serum in PBT. Antibodies were diluted in PBT and tissue was incubated overnight at 4 °C. Antibodies used were anti-cleaved DCP1 (Cell Signaling, CAT 9578S, diluted 1:600), anti-phosphorylated Histone 3 (Cell Signaling, CAT 9701, 1:1000), anti-LbCas12a (Diagenode, CAT C15310263, 1:800), and anti-Smoothened (DSHB, CAT 20C6, 1:200). After serial washes in PBT, tissue was incubated in a secondary antibody (labeled with Alexa Fluorophores, diluted 1:600, Life Technologies) and Hoechst (diluted 1:2000) in PBT for 2 h at room temperature. Following serial washes tissue was mounted in Vectashield (Biozol Diagnostica).

## Imaging

Tissue was imaged on a Leica SP8 laser-scanning confocal microscope with an oil 40x/NA1.4 lens using the sequential scanning mode or a Keyence BZ-X810 fluorescent microscope with an 20x air objective. Image processing and analysis were performed with FIJI. Imaging of adult flies or wings was performed with a Leica M165 FC stereomicroscope equipped with a Leica DFC295 camera.

## Loss of heterozygosity screening

LOH was monitored using heterozygous transgenes encoding fluorescent proteins. Upon LOH of the chromosome arm on which the transgene is located, cells gain or loose fluorescence, which can be visualized using a fluorescence microscope or stereoscope. Plasmids were inserted in attP sites at the distal end of the chromosome, either at cytogenetic position 22 C on chromosome arm 2 L, 59D on arm 2 R, or at 99 F on arm 3 R. Animals were also transgenic for *act5C-Cas12a*$^+$ and HD12aCFD sgRNAs, and LOH was assessed in wing imaginal discs at larval 3rd instar wandering stage. We confirmed that LOH is also readily detected in other tissues, such as the brain and the intestine.

To assay LOH with high resolution the *act5C-mScarlet* plasmid was used, which results in ubiquitous and relative homogenous expression of mScarlet. Third instar larvae were dissected on ice, and wing discs were fixed in 4% paraformaldehyde in PBS at room temperature for 25 minutes. Nuclei were stained with 1 μg/ml Hoechst for 20 minutes, and discs were mounted in Vectashield mounting medium. Imaginal discs were imaged with a Leica SP8 laser-scanning confocal microscope with an oil 40x/NA1.4 lens. Images were analyzed in Fiji by outlining all areas that lost or gained fluorescence and using the measure function to quantify their size. Note that the area that cells with LOH occupy in the tissue reflects both the frequency of LOH (number of cells that gain or lose reporter expression) and the time point of occurrence (number of cell divisions after LOH). While the *act5C-mScarlet* reporter is ideal to observe LOH in isolated tissues by high-resolution imaging, the ubiquitous reporter cannot reliably detect LOH in whole animals, as their multilayered topology obscures LOH events in individual tissues.

For high-throughput LOH screening in living animals, a *GMR11F02-Gal4 UAS-GFP* (for the screen on 2 L) or a *GMR11F02-Gal4 UAS-mScarlet* reporter (for 2 R) was used. *GMR11F02-Gal4* drives specific expression in the pouch region of the wing and haltere imaginal discs. Larvae were immersed in a drop of tap water and observed under a Leica M165 FC fluorescent stereo microscope with 6.3x magnification. To increase consistency, each screen was performed by the same observer (F.P. screened 2 L, while E.P. screened 2 R), who was blinded to larval genotypes. The fluorescent signal from the GMR11F02 reporter in the disc pouch shows natural variation, with stronger signal typically observed at the dorsal edge of the expression domain and weaker fluorescence along the dorsal-ventral boundary. Therefore, the observer focused on larger patches of cells that lost fluorescence and scored the level of LOH per wing disc using the following scale: 0 - no LOH observed; 1 - one or two small clones; 2 - one to three small to large clones; 3 several large clones. The mean LOH score presented in the figures represents the mean score of all wing discs (typically 10 − 20) recorded for a particular genotype.

## Drosophila wing size phenotyping

Dissected *Drosophila* wings were mounted on glass slides and imaged using a Leica M165 FC stereo microscope. Images were used to retrain Cellpose's cyto2 model[61] using human-in-the-loop capabilities and parameters: diameter 140, flow 1, and cell prob. 1. Using this model ('fly_wings') wings were segmented, outlines were exported to Fiji and wing size was measured using the Measure feature. The custom Cellpose model ("Wingpose") is provided in Supplementary Data 4.

## Generation of *trem* alleles

Germline mutations in *trem* were generated as previously described[62]. Briefly, *nos-Gal4VP16 UAS-u$^M$Cas12a*$^+$ flies were crossed to *y w hs-Flp; HD12aCFD0127* sgRNA flies. Male offspring, expressing Cas12a$^+$ in the germline and *trem* specific sgRNAs, were then crossed to *w*$^*$*,+,+* virgin females and offspring harboring mutations in *trem* were identified by genotyping PCR using primers *tremgenofwd1* and *tremgenorev1*. Two mutant alleles were recovered: *trem$^{del10}$* harboring a ten-base pair deletion (CCTGAACAATCCCAG - del10 - GAGCTGCTCCACGAC) and

trem$^{del11}$ (TTGCCTGAACAATCC - del11 - AGGAGCTGCTCCACG). Both shift the reading frame in the first coding exon, resulting in a truncation of about 90% of the coding sequence. Mutant flies were then backcrossed against $w^*,+,+$ animals for five generations to remove potential secondary mutations. Afterwards, a stable stock was generated using a *y w hs-Flp;; MKRS/TM6B* balancer stock. *y w hs-Flp;; trem$^{del10}$/TM6B* and *y w hs-Flp;; trem$^{del11}$/TM6B* stocks both retain the balancer chromosome, as homozygous *trem* mutant animals die in mid larval stage.

### Data analysis and visualization
Data analysis was performed in R using the Tidyverse package. Graphs were created using ggplot2. Microscope images were prepared using Fiji. Figures were created using Affinity Designer V1.

### Statistics and reproducibility
Sample sizes were not predetermined by statistical methods, but are based on established practices in the field to ensure sufficient power to detect phenotypic effects. Experimenters were blinded to the identity of the analyzed genotypes during the LOH screens reported in Fig. 5i and j. No blinding was used during all other experiments. Statistical analysis was performed in R and the details of the statistical tests are given in the figure legends. All experiments, except the large-scale screens reported in Figs. 5i, 5j, 6g, 6h, were repeated multiple times with consistent outcomes, supporting the robustness and reproducibility of the findings.

### Reporting summary
Further information on research design is available in the Nature Portfolio Reporting Summary linked to this article.

## Data availability
All data is contained within the manuscript. Source data are provided with this paper. The 'Wingpose' Cellpose model is provided as Supplementary Data 4. The described plasmids are publicly available from the European Plasmid Repository and Addgene and the fly strains and available from the Vienna Drosophila Resource Center. Source data are provided with this paper.

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

## Acknowledgements

We would like to thank Ryan Loker and Richard Mann (Columbia University, USA) for plasmids and Erich Brunner (University of Zürich, Switzerland) for advice about the construction of the GRACE reporter. Roman Doll (University of Oxford, UK) for discussions and comments on the manuscript. Jianing Zhang, Nikola Knoll, Sophia Schelchshorn, Claudia Strein and Alma Spahic for assistance and discussions. The High-Throughput Sequencing Unit of the Genomics and Proteomics Core Facility and Light Microscopy Facility at DKFZ for support. This work has in part been supported by grants from the European Research Council (DECODE) and the German Research Foundation (SFB1324) to M.B.

## Author contributions

F.P. and M.B. conceived the study. F.P., M.A.B., and J.Z. designed, performed and analyzed experiments. M.S., A.V.B., A.C.M., E.R., A.P., L.G., J.H., L.B.M.K., B.W., M.H., E.P. performed experiments. F.H. designed the sgRNA library and trained the Cellpose model. M.B. acquired funding. F.P. and M.B. supervised the work. F.P. wrote the paper. All authors reviewed and edited the manuscript.

## Funding

## Competing interests

The authors declare no competing interests.
