## [Transparent Peer Review file · Nature Communications]

Improved in vivo gene knockout with high specificity using multiplexed Cas12a sgRNAs

Corresponding Author: Dr Phillip Port

Version 0:

Reviewer comments:

Reviewer #1

(Remarks to the Author)

Knocking out genes via CRISPR-Cas9 catalyzed DNA double-strand breaks (DSBs) is a widely used approach for functional studies of gene function in various biological model systems. The principal advantage of this method is the theoretically unlimited flexibility of targeting any part of the genome that is adjacent to a PAM site, with the optimal outcome mostly being small indels generated by DNA-DSB repair that disrupt the coding sequence. However, this process is limited in practice by sub-optimal sgRNAs, sub-optimal target sequence context and chromatin and transcriptional state of target sequence, to name a few (PMID: 38440123). An additional drawback of CRISPR-Cas9 mediated mutagenesis is the possibility of off-target effects, which are quite difficult to detect (PMID: 36970624), but can create mutations at unintended target sites that may affect functional studies. Still another potential pitfall of knocking-out genes with CRISPR-Cas9 catalyzed DNA DSBs is the potential of un-intended outcomes at the target site, such as large deletions (PMID: 30010673, PMID: 35701408 and PMID: 39496933) and even mega-base scale losses and rearrangements that can truncate large portions of chromosomes, lead to translocations, whole chromosome losses (PMID: 35773341, PMID: 34389729, PMID: 33125898, PMID: 34050011, PMID: 37794590, PMID: 37429857 and PMID: 39386534) and even chromothripsis (PMID: 33846636). Such large genomic rearrangements may result in loss of heterozygosity (LOH) over large regions of the targeted chromosomes (PMID: 37429857 and PMID: 39386534), which also have the potential to create confounding effects for functional studies involving gene knockout. This was reported in a recent study by Lazar et al. (PMID: 38811841), where they showed proximity biases where CRISPR knockouts show unexpected phenotypic similarities to unrelated genes on the same chromosome arm. This "proximity effect" is in fact widespread in CRISPR-Cas9 genetic screens and has led to misappropriation of function to genes in large datasets like DepMap, and Lazar et al. proposed the application of a geometric correction algorithm to potentially remove erroneous assignment of function to genes from large CRISPR-Cas9 screens. In the opinion of this reviewer, "on-target" toxic effects such LOH are as much of a concern with CRISPR-Cas9 or any other DNA DSB mediated mutagenesis application as off-target effects, if not more, because they can affect Megabases of DNA sequences than span over entire chromosome arms and hundreds to thousands of genes.

In this study, Port et al. describe a genome editing strategy that combines an enhanced variant of Cas12a combined with an array of 4 sgRNAs to generate a more efficient gene targeting system than Cas9 based methods. Their experiments demonstrate that their system, referred to in the paper as HD12aCFD, routinely leads to stronger DNA target ablation and more penetrant phenotypes in vivo in live fruit flies compared to a Cas9 based system with 2 sgRNAs per target (referred to as HD_CFD). In addition, the authors assayed for LOH with an extensive number of sgRNA arrays in fruit flies with telomere proximal fluorescent reporters on 2L and 3R, and showed that many of these cause reproducible levels of LOH specifically on the targeted chromosome arm. These LOH events span several Megabase pairs, in agreement with several studies not only in mammalian cells (cited above), but also in *Candida parapsilosis* (PMID: 36416551) and tomato (PMID: 37497643). Finally, the authors showed more experiments that reiterate the higher knock-out efficiency with HDCas12aCFD compared to HD_CFD, and hypothesize a novel role for the gene trade embargo (trem) in wing morphogenesis revealed by the superior efficacy of HDCas12aCFD.

While initial impressions do suggest that HDCas12aCFD creates gene knock-outs with higher efficiency than Cas9 based HD_CFD, the authors need to address some major questions first, particularly regarding extensive LOH, before they can fully establish that HDCas12aCFD is a superior gene knock-out tool. The major critiques are as follows:

1) As stated in the opening paragraph, the ideal outcome for Cas9 or any other DNA DSB based method for gene knock-out

is the generation of small indels around the targeted DNA site in any early coding region. As long as the insertions or deletions do not generate 3 or multiples of 3 base pair alterations, the gene will be knocked out of frame. More importantly, small indels ensure that the genetic effect of DNA sequence modification is restricted to the target gene and the target gene alone, loss of larger amounts of DNA has the potential to affect neighbouring genes, which then calls into question whether the phenotypic effect is from knocking out the targeted gene or any another gene that was also affected by a much larger deletion. The author's extensive experiments with the LOH reporters unequivocally demonstrate that extensive LOH is a common and in some cases frequent (~15-20%, Figure 4d) outcome of several HDCas12CFD arrays. While there is no further characterization of whether the LOH is a consequence of mitotic recombination (copy neutral) or deletions/chromothripsis (copy loss), both outcomes are problematic for gene knockouts. Homologous recombination (HR) associated with crossing over could result in LOH, which could lead to a buildup of recessive alleles downstream of the recombination site. However, gene conversion associated with HR would also repair the broken DNA region on the targeted chromosome with DNA sequences from the homologous donor allele and thereby restore function, unless the donor allele was also mutated. This outcome would therefore suppress gene knock-outs. On the other hand, deletions would lead to the same type of "proximity" effects reported by Lazar et al. (PMID: 38811841) for Cas9 based mutagenesis screens, as multiple genes will lose at least one allele downstream of the DSB if only one homologue arm was affected, and will be also knocked out if both homologues are affected.

The authors contend that LOH in drosophila occurs primarily through mitotic recombination. However, one of the two papers they cited, Brunner et al. (PMID: 31196871) characterized a random assortment of 57 events from a cohort of 253 recombination events induced by a Cas9 DSB, and reported that 19 contained an exchange of flanking DNA from both homologues without any indels, 2 contained exchanges with indels, and the remaining 36 of 57 (~63%), which are a majority of these "recombination" events, were deletions that had lost regions of the chromosome distal to the DSB. Therefore, the foremost concern regarding this study is that the primary source of the stronger phenotypic effects with HDCas12aCFD compared to HD_CFD is due to an increased incidence of extensive LOH that occurs via deletions, which then potentially nullifies the utility of this approach as genotype to phenotype conclusions cannot be made if multiple genes are routinely knocked out at significant frequencies (~15-20%, Figure 4d). This possibility has not been addressed by the authors in any part of the manuscript. The authors performed Sanger sequencing around the DSB site for three HDCas12aCFD arrays (0002, 0324 and 0476). As no local sequence effects were observed for the low LOH score arrays 0324 and 0476, it may suggest DNA modifications at target sites by low efficiency arrays cannot be scored by Sanger sequencing. However, for the high LOH score array 0002, sequence changes were observed around the DSB site, therefore local indel frequencies can be estimated from algorithms such as TIDE (<http://shinyapps.org/datacurators.nl/tide/>) and ICE (<https://ice.editco.bio/#/>). And than this frequency could be compared to the % area of reporter loss on wing discs as a measure of LOH frequency, to then compare frequencies of local deletions versus extensive LOH. And while it is true that even standard Cas9 based mutagenesis approaches lead to extensive LOH from deletions (references provided earlier), the study by Regan et al. (PMID: 39386534) in mouse embryonic stem cells reported that for one of their extensively characterized Cas9-gRNA sites, LOH comprised of 5% of all events, which would constitute a minor fraction of Cas9 DSB repair outcomes compared to the ~85.5% that had indels as characterized by Sanger sequencing around the DSB site followed by analysis with ICE. This reviewer suggests the following analyses to address the contribution of local sequence changes versus extensive LOH:

- a) The authors have to perform more Sanger sequencing based analyses around the sgRNA array sites from a random assortment of their high LOH arrays and compare that to the absolute % of LOH they observed in wing discs. They must then perform the same analysis with Cas9 arrays via HD_CFD, ideally around the same genetic loci, and only then can we evaluate if the increased knock out efficiency of the HDCas12a array is from an increase in local sequence changes versus extensive DNA loss. These experiments are strongly recommended.
- b) The authors assayed a number of sgRNA arrays for their effect on wing size and showed the effect of expected positive and negative controls met expectations. However, they also saw an effect on wing size from a host of genes not known to affect this process, and there could be proximity effects in this dataset. Therefore, the authors should replot the data in Supplementary figure 7a, reorganizing the list of genes on the X-axis via their relative chromosome positions as opposed to effect size, and this could reveal if knocking out genes that are in the vicinity of and centromere proximal to genes that are known regulators of wing size also show proximity-based effects. This analysis is also strongly recommended.
- c) Characterizing some of the LOH events, to determine if they are from reciprocal chromosome exchanges or deletions would also go a long way to alleviate the concern that large deletions are significantly affecting observed phenotypes. This analysis, however, is likely to be very intensive and time-consuming, and may not even be feasible, even though these characterizations would be very convincing one way or another.

2) Another consequence of the occurrence of extensive LOH is they bring into question several sections of the paper. These are as follows:

- a. The entire section entitled "Mutations caused by multiplexed editing are locally restricted" along with figure 3 is rendered moot by the next sections that shows that extensive LOH is detected for many sgRNA arrays. Moreover, if the results in Figure 3 are reevaluated after acknowledging that chromosomal sequences distal from the centromere and towards the telomere are often deleted following DSBs induced with sgRNA arrays, it adequately explains why sgRNA arrays centromere proximal to *se* at *GstO3* affect its function but not at the telomere proximal *Ra* locus, when both arrays are practically equidistant from the beginning and end of *se* coding sequences respectively and why the telomere proximal array at CG5892 has no effect on the centromere proximal *ebony* locus. Internal sgRNAs at *forked* showed no effect, but this may reflect that *forked* is essential to viability in fruit flies and therefore large deletions are lost. Therefore, this entire section should be removed, and could potentially be replaced with a further analysis of the deletion alleles generated at the target sites by sgRNA arrays used in actual gene knock-out studies, similar to the analysis in Figure 1C.
- b. The section that utilizes LOH to determine off-target effects of sgRNA arrays along with experiments in Figure 4 are unfortunately ill-conceived. The authors contend that a lack of off target LOH is evidence for lack of all off-target effects from sgRNA arrays. However, typical repair outcomes for Cas9 DSBs are mostly small indels, therefore off-target effects are also similarly expected to mostly induce small indels. And it is important to acknowledge that the off-target sites of the sgRNA

arrays may not necessarily be clustered like the on-target site, therefore even if LOH outcomes are as frequent as localized indels on the sgRNA target sites, the same may not be true for off-target sites. Off-target sites also may not be reproducibly cleaved, and thus the non-reproducible LOH observed could also be from off-target cleavage instead of spontaneous events. To compare this to a recent study by Höijer et al. in Zebra fish (PMID: 35110541) that characterized off-target repair outcomes at predicted off-target sites, 76% were small deletions, 4% large deletions (>50bps) and 25% were small insertions. Even allowing for differences in repair outcomes between fruit flies and zebra fish and for Cas9 and Cas12a DSBs, nevertheless the LOH assay alone cannot account for the presence or absence of small indels and therefore cannot eliminate the possibility of off-target effects. Unless the authors perform Illumina sequencing or other NGS on bulk DNA from flies that express a select few HD12aCFD arrays at multiple predicted off-target sites for all four individual sgRNAs, the LOH analysis alone does not eliminate the possibility of off-target effects.

c. The final two sections entitled “Multiplexed Cas12a+ gene targeting outperforms established Cas9-based methods” and “Increased gene disruption efficiency reveals a somatic function for trade embargo” are also complicated by the potential for LOH. Data in Figure 5 will be greatly strengthened if LOH could also be assayed at any of the three genes targeted, if there is a genetically assayable marker telomere proximal to these respective genes. And the novel function of trem may also be a consequence of LOH mediated knock out of a gene telomere proximal to trem on 3R, but this is a difficult question to address. Unfortunately, the two characterized mutants of trem with identified indels both proved to be homozygous lethal, and thus the wing development function could not be corroborated. Nevertheless, if the experiments previously suggested show LOH frequency is comparable to local deletions for high efficiency sgRNAs, the assertions in Figure 5 need to be qualified by stating a lot of the cells with gene knock-outs potentially contain chromosome truncations. And those in Figure 6 regarding a novel wing development function for trem ought to be removed entirely, if the possibility of a confounding effect due to frequent LOH cannot be eliminated.

To reiterate, the major concerns regarding this manuscript involve experiments that address the frequency of LOH and its potentially confounding effects on other assertions made in the manuscript, without which publication cannot be recommended. To summarize in a sentence, if the increase in gene knockouts from HD12aCFD is primarily due to an increase in extensive LOH, then this basically nullifies the utility of this approach compared to conventional approaches that use one or two DSBs, instead of 4 clustered DSBs. Finally, a minor concern accompanying this is that manuscript itself ought to be re-written, to acknowledge the potentially confounding effects LOH can have for gene knock-out studies. Also, the reference in the legend for Figure 4 should refer to extended data Figure 3. Not 2.

Reviewer #2

(Remarks to the Author)

In this manuscript by Port et al., the authors describe a highly efficient gene-targeting system in *Drosophila* that leverages multiplexed Cas12a sgRNAs. By utilizing quadruple sgRNA arrays alongside an enhanced Cas12a variant (Cas12a+), they demonstrate the system's ability to efficiently mutate targeted genes. The authors suggest that this approach offers improved knockout efficiency compared to traditional Cas9-based methods, revealing phenotypes that previous techniques may have missed.

The strength of this study is that it expands the CRISPR toolbox in *Drosophila* to Cas12a and describes effective strategies for constructing Cas12a/sgRNA reagents for tissue-specific gene editing. Using this strategy, the authors generated a small library of HD12aCFD sgRNA arrays targeting 241 genes, which can be a useful resource for the *Drosophila* community. In addition, the authors designed in vivo assays to test several other properties associated with CRISPR mutators that have not been commonly examined in *Drosophila*, including DSB-induced toxicity, DNA disruptions beyond the target sites, and off-target effects (even though that was limited to a specific chromosomal segment).

Although most of the experiments were nicely done and the results were mostly convincing (with the caveat of missing some technical information that is important for evaluation), the reviewer thinks that the authors have overstated the advantage of the Cas12a system over Cas9 systems. The authors' claims were based on unfair comparisons involving Cas9 sgRNAs with suboptimal designs and unequal numbers of sgRNAs for the two systems. Some caveats were presented as being unique to Cas9 systems but are in fact common to all CRISPR systems. Thus, the Cas12a system appears to be oversold while the strengths of the Cas9 system were understated.

Major concerns

1. A major conclusion of this study was that the Cas12a+ and HD12aCFD system is superior to Cas9 systems in gene mutagenesis (“gene targeting with Cas12a+ and multiplexed sgRNAs achieves superior gene disruption efficiency compared to established Cas9-based systems across diverse tissues”). However, this very general conclusion was based on incomplete/biased comparisons. The efficiency of CRISPR mutagenesis is affected by multiple factors, including the choice of targeting sites, gRNA scaffold sequence, the number of gRNAs used, and Cas protein expression level and temporal patterns. In the comparison of Cas9 and Cas12a systems in this study, even if the Cas9 and Cas12a have comparable expression (of which the technical information is missing), the influence of gRNA target sites, numbers, and gRNA scaffold were not sufficiently considered. Numerous studies have shown that the gRNA target sequences have a large impact on editing efficiency (e.g. DOI: 10.1038/nmeth.3543; DOI: 10.1038/nbt.3437), but the manuscript did not discuss what criteria were used to select the target sequences for either system and what measures were taken to ensure that they are efficient. More importantly, studies have shown that optimizing the gRNA scaffold sequence for Cas9 can dramatically increase Cas9 activity (DOI: 10.1016/j.cell.2013.12.001; DOI: 10.1126/science.aao0932). Further work in *Drosophila* has incorporated these optimizations and demonstrated their improved efficiencies over the original scaffold (DOI: 10.1534/genetics.118.301736; DOI: 10.1073/pnas.2014255118). However, the authors have completely ignored these improvements in this manuscript, and the Cas9 sgRNAs examined in this study seem to still use the original suboptimal

scaffold. Thus, although the results in this study suggest that the new HD12aCFD system is better than the authors' own HD_CFD system, they cannot be generalized for claiming that the Cas12a+ system is better than Cas9 systems. If this is the authors' intention, they should compare HD12aCFD to similar Cas9 systems with the optimized gRNA scaffold and to make sure that the efficiency and the number of target sites for each gene tested are comparable for both systems.

2. The authors claimed that "Cas12a+ activity can be effectively controlled in time and space, as phenotypic changes remained confined to the targeted regions - a precision that has been challenging to achieve with Cas9". However, this point was not convincingly shown. The Cas12a+ expression in this study was controlled and determined by the Gal4 lines used, not by some intrinsic differences between Cas12a+ and Cas9. Similar Gal4/UAS strategies can be used to control the tissue-specificity of Cas9 as well. To support this claim, the authors need to present data demonstrating that Cas12a+ is intrinsically more specific than Cas9.

3. To show that gene disruptions caused by HD12aCFD sgRNA arrays are locally restricted, the authors checked phenotypes associated with several recessive genes (Figure 3). However, to see the recessive phenotypes, both alleles of the tested genes need to be mutated. Thus, the lack of a phenotype does not necessarily correlate with the lack of DNA disruption, and the results do not truly reflect the frequency of broader disruptions at the targeted loci.

4. In somatic targeting of the STAR reporter, if the editing outcomes in the majority of the cells are full deletions of the sequences between the first and last targeting sites, one would expect to see a relatively distinct PCR band corresponding to such deletions (because shorter amplicons are preferred products in genomic PCR). However, Figure 1b does not show this. The results instead suggest that full deletions are rare or at least low in frequency. The germline targeting results in Figure 1c cannot be used to infer somatic targeting, as DNA repair can differ significantly between the soma and the germline. While I agree that the results suggest that deletions (that can be between any two target sites) become more frequent with more target sites overall, the writing (especially by describing the results of somatic and germline data in the same sentence) can leave people a false impression about the frequency of large deletions in the soma.

5. In Figure 1a, how would the editing outcome be different if only the most distal target sites were available? In other words, how would the distance between two target sites affect the editing outcome?

6. If the LOH events in the screening were truly due to background stochastic mitotic recombination, they could be observable even in the control experiment where Cas12a or sgRNAs were not expressed. Was this the case?

7. Some important technical information is missing for some experiments. It is difficult to evaluate the results without the information. Below are some examples.

- In Figures 1d, 3, and 5, where were Cas9 and Cas12a expressed? What measures were taken to ensure that their expression levels were comparable? How were gRNAs expressed (by UAS or a ubiquitous promoter)? What was done to ensure that the gRNAs were expressed at comparable levels?
- In Figure 5, how was inducible mutagenesis of Notch, neur, and smo done?
- In Figure 1c, how many sequences are presented for each genotype/batch? How decisions were made regarding which sequences are included in the figure panel?
- The manuscript did not talk about how gRNA target sites were selected and how to make sure that they would be efficient.
- Scoring of the LOH index was not clearly defined.

Minor concerns

1. "CRISPR editing often produces genetics mosaics". This sentence reads as if genetic mosaicism is avoidable. But in tissue-specific CRISPR experiments, DNA DSBs are generated and repaired independently in each cell in the manipulated tissue. The outcome should always be genetic mosaics.

2. The authors claimed that "...for Cas9, the most widely used CRISPR nuclease, creating multiplexed arrays of more than two sgRNAs remains technically challenging due to their large size and repetitive nature." It's true that gRNAs for Cas9 are larger than those for Cas12a, but multiplexed Cas12a gRNAs also contain repetitive sequences. Several studies in recent years (including those published by some of the same authors) have already reported convenient methods for making multiplexed (e.g. 2-6) sgRNAs for Cas9 (REF). So, I am not entirely convinced by the authors' argument that the Cas12a system is much better than Cas9 in this regard.

3. Not all results in Supplementary Fig. 5 were described in the main text. Although some details were provided in the figure legend, the results should be at least briefly described in the main text.

4. Supplementary tables were not provided.

Reviewer #3

(Remarks to the Author)

Reviewer #4

(Remarks to the Author)

Port et.al. present a Cas12a-based multiplexed system that targets each gene with four sgRNAs to enhance editing efficiency, using *Drosophila* as an in vivo model. Their findings demonstrate that the synergistic action of multiple sgRNAs leads to efficient deletion formation, significantly increasing the proportion of loss-of-function mutations. To rigorously evaluate off-target effects, they introduce a novel screening assay that enables real-time visualization of CRISPR-induced chromosomal alterations in living organisms and demonstrated no detectable off-target effects. Furthermore, they reveal the key roles of *trem* in wing development by using multiplexed Cas12a system. Overall, the study is intriguing and most of the data support their conclusion. My major concern is that the Cas12a multiplexed system in *Drosophila* has been described by the same group but there is no clear description that what the improvement of the system is and there is no data to support this system is outperform than previous one.

Specific comments:

1. Figure 1a illustrates the report structure targeted by the Cas12a system, where the target sites are identical. However, there is no description of the structure of the multiplexed sgRNAs array. Are the same sgRNAs used to target these identical sites, or is the array multiplexed with various sgRNAs? Including a diagram to depict the Cas12a sgRNAs system would be helpful. Additionally, if the array reporter could be designed to contain different target sites and be targeted by six distinct sgRNAs (with similar editing efficacy), it would provide further insights into assessing editing efficacy.
2. What is the x-axis in Figure 1c? How can the non-linear changes in deletion percentage among arrays be explained? For instance, why does the 2x array show a higher percentage of deletion than the 3x array in the first region? Additionally, while the 4x array exhibits the highest deletion percentage, there is a decline in the 5x and 6x arrays. Clarification on these trends would be helpful.
3. In Figure 1d, the authors compared the Cas12a multiplexed system with the Cas9 system. To draw a solid conclusion that Cas12a outperforms Cas9, it would be helpful to include the deletion efficacy of each single guide used in Figure 1d for both Cas12a and Cas9. As well as showing the deletion efficacy of Cas12a with two guides. Also, in figure 5 and 6.
4. Although, the claim 'the use of multiple sgRNAs creates functional redundancy' has been demonstrated by previous studies, the data in Figure 1 does not provide enough support for this claim. The authors could strengthen this point by citing relevant literature or including additional data, such as single sgRNAs efficacy controls, to better support their argument.
5. Please explain the reason that the mutation test in figure 3 only use up to 2 guides instead of 4 guides.
6. The authors claim that off-target effects of Cas12a editing are undetectable. Is this due to the limitations of the detection system? Whole-genome sequencing would be useful to determine whether low-frequency off-target effects are present. Does these low off-target effects are universal in high order model or only in fly model. For future applications, in vivo off-targets information in organoid or mice model are useful (organoid or mice).
7. Statistics is missing in figures 6b, 6c, and 6d.

Version 1:

Reviewer comments:

Reviewer #1

(Remarks to the Author)

Following on from the initial review of the manuscript, the primary concern raised by this reviewer was the possibility of chromosome-arm scale deletions induced by clustered Cas12a DSBs leading to copy number-loss Loss of Heterozygosity (CL-LOH), which would produce confounding proximity based phenotypic effects and limit the efficacy of this approach as a novel gene knockout strategy. Upon reading the revised manuscript, this reviewer commends the authors for embracing this concern and going above and beyond to address it. The experiments in the revised manuscript thoroughly demonstrating an absence of proximity based phenotypic effects from gene knockouts, and the presence of copy number gain events strongly support a predominance of DSB induced CN-LOH and a relative absence of DSB induced CL-LOH events. Moreover, the authors also produced a supporting figure showing that RNAi mediated *trem* knockdown also shows a similar, albeit weaker wing phenotype as the HD Cas12aCFD mediated *trem* knockout, which ameliorate concerns regarding the novel proposed function of *trem*. This reviewer therefore recommends publication, notwithstanding a few minor comments that can be addressed with a few small changes in the manuscript text, which are as follows:

- 1) The copy-number neutral LOH (CN-LOH) events in the paper are ubiquitously described as mitotic recombination events. While it is true that allele specific Cas9 DSBs induce substantial levels of mitotic recombination associated with crossing over in *Drosophila* (PMID: 33750946), an important difference is that DSBs induced in this paper produce breaks on both homologues. This raises the formal possibility that CN-LOH events instead occur via reciprocal end-joining events, which have been demonstrated in human HT1080 cells, a p53-positive fibrosarcoma cell line (PMID: 32873648). The reviewer does acknowledge that frequent mitotic recombination has been demonstrated before in *Drosophila*, nevertheless acknowledging the possibility of reciprocal end-joining events covers all bases.
- 2) The method of tracking LOH on chromosome arms with integrated fluorescent reporters to detect off-target genome editing is indeed very convenient, but this approach is limited to tracking events only on the marked chromosome arms. This caveat is usually not acknowledged in the text, rather the phrase "off-target activity" is used without any qualifiers. It is therefore recommended that the text in lines 91 and 420 be changed to "off-target activity on the two chromosome arms with LOH reporters".
- 3) Figure 3 legend line 244 refers to supplementary figure 1, when it seems it should refer to supplementary figure 2.

Reviewer #2

(Remarks to the Author)

The authors have done a very nice job revising the manuscript by adding substantial new results. These new results and revisions of text addressed my previous concerns. The revised manuscript is a very nice piece of work which represents a significant advancement in CRISPR technology and application in *Drosophila*. I am supportive of its publication by Nature Communications but with two minor comments.

1. Figure 2 characterized Cas12a+ activity using the GRACE reporter. UAS-Cas12a+ appears to have low levels of leaky activity in the wing disc. How is it compared to the leaky activities of previous UAS-Cas9 strains generated by the same group? This information would be useful for people deciding between the Cas9 and Cas12a+ system.

2. It is not clear how the sequencing results in Supplementary Figure 6 were generated. Were they derived from PCR from a single animal or a mixture of multiple larvae? Or were they from dissected larval wing discs?

Reviewer #3

(Remarks to the Author)

Reviewer #4

(Remarks to the Author)

The authors have addressed most of my comments. I don't have any further questions.

Point-by-Point Response to Referees

We thank all four reviewers for their constructive and insightful feedback on our manuscript. The points you raised have been addressed through extensive additional experimental work, as detailed in our responses below. We believe these revisions have substantially strengthened the study and hope you will agree that the manuscript is now suitable for publication.

All original reviewer comments are copied below with our replies in bold.

Reviewer #1 (Remarks to the Author):

Knocking out genes via CRISPR-Cas9 catalyzed DNA double-strand breaks (DSBs) is a widely used approach for functional studies of gene function in various biological model systems. The principal advantage of this method is the theoretically unlimited flexibility of targeting any part of the genome that is adjacent to a PAM site, with the optimal outcome mostly being small indels generated by DNA-DSB repair that disrupt the coding sequence. However, this process is limited in practice by sub-optimal sgRNAs, sub-optimal target sequence context and chromatin and transcriptional state of target sequence, to name a few (PMID: 38440123). An additional drawback of CRISPR-Cas9 mediated mutagenesis is the possibility of off-target effects, which are quite difficult to detect (PMID: 36970624), but can create mutations at unintended target sites that may affect functional studies. Still another potential pitfall of knocking-out genes with CRISPR-Cas9 catalyzed DNA DSBs is the potential of un-intended outcomes at the target site, such as large deletions (PMID: 30010673, PMID: 35701408 and PMID: 39496933) and even mega-base scale losses and rearrangements that can truncate large portions of chromosomes, lead to translocations, whole chromosome losses (PMID: 35773341, PMID: 34389729, PMID: 33125898, PMID: 34050011, PMID: 37794590, PMID: 37429857 and PMID: 39386534) and even chromothripsis (PMID: 33846636). Such large genomic rearrangements may result in loss of heterozygosity (LOH) over large regions of the targeted chromosomes (PMID: 37429857 and PMID: 39386534), which also have the potential to create confounding effects for functional studies involving gene knockout. This was reported in a recent study by Lazar et al. (PMID: 38811841), where they showed proximity biases where CRISPR knockouts show unexpected phenotypic similarities to unrelated genes on the same chromosome arm. This “proximity effect” is in fact widespread in CRISPR-Cas9 genetic screens and has led to misappropriation of function to genes in large datasets like DepMap, and Lazar et al. proposed the application of a geometric correction algorithm to potentially remove erroneous assignment of function to genes from large CRISPR-Cas9 screens. In the opinion of this reviewer, “on-target” toxic effects such LOH are as much of a concern with CRISPR-Cas9 or any other DNA DSB mediated mutagenesis application as off-target effects, if not more, because they can affect Megabases of DNA sequences than span over entire chromosome arms and hundreds to thousands of genes.

In this study, Port et al. describe a genome editing strategy that combines an enhanced variant of Cas12a combined with an array of 4 sgRNAs to generate a more efficient gene targeting system than Cas9 based methods. Their experiments demonstrate that their system, referred to in the paper as HD12aCFD, routinely leads to stronger DNA target ablation and more penetrant phenotypes in

vivo in live fruit flies compared to a Cas9 based system with 2 sgRNAs per target (referred to as HD_CFD). In addition, the authors assayed for LOH with an extensive number of sgRNA arrays in fruit flies with telomere proximal fluorescent reporters on 2L and 3R, and showed that many of these cause reproducible levels of LOH specifically on the targeted chromosome arm. These LOH events span several Megabase pairs, in agreement with several studies not only in mammalian cells (cited above), but also in *Candida parapsilosis* (PMID: 36416551) and tomato (PMID: 37497643). Finally, the authors showed more experiments that reiterate the higher knock-out efficiency with HDCas12aCFD compared to HD_CFD, and hypothesize a novel role for the gene trade embargo (*trem*) in wing morphogenesis revealed by the superior efficacy of HDCas12aCFD.

While initial impressions do suggest that HDCas12aCFD creates gene knock-outs with higher efficiency than Cas9 based HD_CFD, the authors need to address some major questions first, particularly regarding extensive LOH, before they can fully establish that HDCas12aCFD is a superior gene knock-out tool.

We thank the reviewer for their thorough and detailed feedback. The reviewer raises an important concern: that clustered double-strand breaks (DSBs) could induce large deletions or chromosomal truncations, potentially explaining the strong phenotypes we observe with HD12aCFD. This concern is well-founded given extensive data from mammalian systems. However, whether such effects also occur with relevant frequencies in *Drosophila* has not been previously investigated.

To address this question, we performed multiple additional experiments that provide several new lines of evidence. These data demonstrate that copy-number neutral loss of heterozygosity (LOH) is the dominant mechanism in *Drosophila*, and that the enhanced phenotypes with multiplexed Cas12a mutagenesis are not caused by proximity effects. Specifically, our new experiments show: (i) quantitative analysis of cells that gained or lost a LOH reporter reveals that copy-number neutral LOH predominates in flies; (ii) extended analysis of genes neighboring those with known phenotypes does not support widespread proximity effects; (iii) phenotypes are not clustered along chromosome arms, with no bias toward stronger phenotypes near centromeres, further arguing against proximity effects; and (iv) established differences in chromosome biology between *Drosophila* and mammalian cells provide a mechanistic explanation for these observations. We detail these experiments and supporting evidence below.

The major critiques are as follows:

1) As stated in the opening paragraph, the ideal outcome for Cas9 or any other DNA DSB based method for gene knock-out is the generation of small indels around the targeted DNA site in any early coding region. As long as the insertions or deletions do not generate 3 or multiples of 3 base pair alterations, the gene will be knocked out of frame. More importantly, small indels ensure that the genetic effect of DNA sequence modification is restricted to the target gene and the target gene alone, loss of larger amounts of DNA has the potential to affect neighbouring genes, which then calls into question whether the phenotypic effect is from knocking out the targeted gene or any another gene that was also affected by a much larger deletion. The author's extensive experiments with the LOH reporters unequivocally demonstrate that extensive LOH is a common and in some cases frequent (~15-20%, Figure 4d) outcome of several HDCas12CFD arrays. While there is no further

characterization of whether the LOH is a consequence of mitotic recombination (copy neutral) or deletions/chromothripsis (copy loss), both outcomes are problematic for gene knockouts.

We have now analyzed whether the frequent LOH events we observe result from mitotic recombination or deletion/chromothripsis. These mechanisms can be distinguished by their molecular outcomes: mitotic recombination produces copy-number neutral LOH with balanced proportions of cells gaining or losing a particular chromosome arm, whereas deletions or chromothripsis result only in loss of genetic material without compensating gains.

To determine the relative contribution of each mechanism, we quantified the ratio of chromosome arm gain versus loss events using a heterozygous reporter gene. We performed this analysis with five different HD12aCFD sgRNA arrays, two of which cut in close proximity to the reporter. The results, presented in the new Figure 5c and new Supplemental Figure 5, demonstrate that cells with gained or lost reporter signals are present in approximately equal ratios. This finding is consistent with mitotic recombination as the dominant form of LOH and argues against a significant role for extensive deletions affecting entire chromosome arms, as reported in mammalian systems (e.g., Lazar et al.).

Importantly, established differences in chromosome biology provide a mechanistic basis for the distinct behavior between *Drosophila* (and other Diptera) and mammalian systems. Unlike mammalian cells, *Drosophila* somatic cells maintain homologous chromosome pairing in somatic cells throughout development (e.g. PMIDs 31582744, 38722810), positioning homologs in close proximity and greatly increasing the likelihood of inter-homolog exchange following DNA double-strand breaks. We have incorporated this mechanistic explanation into the Discussion of the revised manuscript (Lines 600-607).

Homologous recombination (HR) associated with crossing over could result in LOH, which could lead to a buildup of recessive alleles downstream of the recombination site. However, gene conversion associated with HR would also repair the broken DNA region on the targeted chromosome with DNA sequences from the homologous donor allele and thereby restore function, unless the donor allele was also mutated. This outcome would therefore suppress gene knock-outs.

We agree that repair of DSBs through homologous recombination is likely to restore the original sequence at the cut site. However, our system maintains Cas nuclease and sgRNA expression for extended periods, enabling repeated targeting of restored sites. Consequently, cells that initially undergo copy-number neutral LOH have a high probability of subsequent mutagenesis at the sgRNA target site, as repeated DSBs will eventually be repaired through error-prone non-homologous end joining.

We also acknowledge that mitotic recombination leads to homozygosity of recessive alleles, potentially causing undesired phenotypes. However, experimentally induced LOH for mosaic analysis represents one of the most widely used genetic perturbation methods in *Drosophila* research (the foundational references PMIDs 8404527, 11311363, 1628809 have been collectively cited >3000 times), which has enabled numerous breakthrough discoveries in developmental biology and genetics. This extensive experimental precedent demonstrates that in *Drosophila*, the risk from accumulating recessive mutations is manageable and that methods inducing frequent copy-number neutral LOH are compatible with meaningful

biological investigations. The robust track record of LOH-based approaches in this model organism therefore supports the validity of our experimental strategy. We now discuss this important context in the revised manuscript (Lines 613-617).

On the other hand, deletions would lead to the same type of “proximity” effects reported by Lazar et al. (PMID: 38811841) for Cas9 based mutagenesis screens, as multiple genes will lose at least one allele downstream of the DSB if only one homologue arm was affected, and will be also knocked out if both homologues are affected.

The authors contend that LOH in *Drosophila* occurs primarily through mitotic recombination. However, one of the two papers they cited, Brunner et al. (PMID: 31196871) characterized a random assortment of 57 events from a cohort of 253 recombination events induced by a Cas9 DSB, and reported that 19 contained an exchange of flanking DNA from both homologues without any indels, 2 contained exchanges with indels, and the remaining 36 of 57 (~63%), which are a majority of these “recombination” events, were deletions that had lost regions of the chromosome distal to the DSB.

The cited experiment by Brunner et al. was performed on the 4th chromosome of *Drosophila*, which presents important limitations for comparison to our study. The 4th chromosome, which encodes only ~0.5% of protein-coding genes, is a specialized chromosome that lacks recombination between homologs (PMIDs 28426351, 30401762), making it fundamentally different from the major chromosome arms. Findings regarding recombination frequencies on this chromosome therefore cannot be extrapolated to the bulk of the fly genome. Additionally, the cited experiment showed a total recombination rate of only ~3% (253 of 8,604 flies), therefore the absolute frequency of deletion alleles in this special szenario is below 2%.

We consider the study by Allen et al. (PMID 33444322) more relevant to our work, as these authors analyzed multiple loci across the major chromosomes. Critically, this study presents extensive imaging data from imaginal discs following CRISPR-induced LOH (Figures 1, 2, 3, and 5), consistently showing approximately balanced levels of copy number gain and loss events. This pattern mirrors our results described above and strongly supports mitotic recombination, rather than deletions, as the predominant LOH mechanism. Importantly, Allen et al. used Cas9 rather than Cas12a, indicating that the balanced gain/loss pattern we observe is not specific to Cas12a-mediated double-strand breaks but represents a general feature of CRISPR-induced LOH in *Drosophila* somatic tissues.

Therefore, the foremost concern regarding this study is that the primary source of the stronger phenotypic effects with HDCas12aCFD compared to HD_CFD is due to an increased incidence of extensive LOH that occurs via deletions, which then potentially nullifies the utility of this approach as genotype to phenotype conclusions cannot be made if multiple genes are routinely knocked out at significant frequencies (~15-20%, Figure 4d). This possibility has not been addressed by the authors in any part of the manuscript. The authors performed Sanger sequencing around the DSB site for three HDCas12aCFD arrays (0002, 0324 and 0476). As no local sequence effects were observed for the low LOH score arrays 0324 and 0476, it may suggest DNA modifications at target sites by low efficiency arrays cannot be scored by Sanger sequencing. However, for the high LOH score array 0002, sequence changes were observed around the DSB site, therefore local indel frequencies can

be estimated from algorithms such as TIDE (<http://shinyapps.datacurators.nl/tide/>) and ICE (<https://ice.editco.bio/#/>). And then this frequency could be compared to the % area of reporter loss on wing discs as a measure of LOH frequency, to then compare frequencies of local deletions versus extensive LOH. And while it is true that even standard Cas9 based mutagenesis approaches lead to extensive LOH from deletions (references provided earlier), the study by Regan et al. (PMID: 39386534) in mouse embryonic stem cells reported that for one of their extensively characterized Cas9-gRNA sites, LOH comprised of 5% of all events, which would constitute a minor fraction of Cas9 DSB repair outcomes compared to the ~85.5% that had indels as characterized by Sanger sequencing around the DSB site followed by analysis with ICE. This reviewer suggests the following analyses to address the contribution of local sequence changes versus extensive LOH:

a) The authors have to perform more Sanger sequencing based analyses around the sgRNA array sites from a random assortment of their high LOH arrays and compare that to the absolute % of LOH they observed in wing discs. They must then perform the same analysis with Cas9 arrays via HD_CFD, ideally around the same genetic loci, and only then can we evaluate if the increased knock out efficiency of the HD Cas12a array is from an increase in local sequence changes versus extensive DNA loss. These experiments are strongly recommended.

While we appreciate this experimental suggestion, we believe that correlating Sanger sequencing of PCR amplicons with LOH frequency has significant technical limitations that would preclude definitive conclusions about large deletions and their impact on knockout efficiency. Most critically, extensive deletions at target sites cannot be detected by PCR amplification, as deletion of primer binding sites prevents amplification entirely. Additionally, PCR exhibits inherent bias toward shorter products, causing local deletions to be overrepresented in sequencing data. These factors create a systematically biased view of on-target editing outcomes and severely limit quantitative assessment, particularly when deletions are prevalent.

One might argue that samples showing detectable LOH but no apparent on-target mutations by Sanger sequencing (as presented in Supplementary Fig. 6) could indicate large deletions. However, this interpretation overlooks a key mechanistic consideration: in situations with limited cutting activity mitotic recombination might restore the original target site without subsequent re-cutting and mutagenesis. Therefore, under these conditions the combination of detectable LOH with undetectable on-target mutations is equally consistent with copy-number neutral recombination.

Given these technical constraints, we believe the experimental approaches we describe in our revised manuscript provide more robust evidence to address the reviewer's central question: whether the enhanced phenotypes observed with HD12aCFD result from extensive deletions or reflect more efficient knockout of target genes. Importantly, while our experiments tackle this question from different angles, they uniformly point to efficient target gene deletion rather than proximity effects as causative for the observed phenotypes.

b) The authors assayed a number of sgRNA arrays for their effect on wing size and showed the effect of expected positive and negative controls met expectations. However, they also saw an effect on wing size from a host of genes not known to affect this process, and there could be proximity effects in this dataset. Therefore, the authors should replot the data in Supplementary figure 7a,

reorganizing the list of genes on the X-axis via their relative chromosome positions as opposed to effect size, and this could reveal if knocking out genes that are in the vicinity of and centromere proximal to genes that are known regulators of wing size also show proximity-based effects. This analysis is also strongly recommended.

This has been an excellent suggestion, and we have now performed this analysis as shown in the new Supplemental Figure 3. Using the karyoploteR R package, we accurately mapped wing size phenotypes obtained with different HD12aCFD sgRNA arrays relative to the genomic position of their target gene. This analysis was facilitated by our extensive perturbation dataset (which we substantially expanded during the revisions and now comprises 387 genes) and the compact organization of the *Drosophila* genome, with the vast majority of genes located on just five major chromosome arms.

In contrast to human cells, where perturbation effects increase in strength when targeting genes closer to centromeres (e.g., Figure 1 in Lazar et al.), we observe no such positional bias in our dataset. Genes causing wing size reduction are distributed randomly along chromosome arms, with no enrichment near centromeres or other chromosomal landmarks. This absence of proximity-dependent effects provides strong evidence that the mechanisms driving extensive deletions in human cells do not operate at significant frequencies in the *Drosophila* system.

c) Characterizing some of the LOH events, to determine if they are from reciprocal chromosome exchanges or deletions would also go a long way to alleviate the concern that large deletions are significantly affecting observed phenotypes. This analysis, however, is likely to be very intensive and time-consuming, and may not even be feasible, even though these characterizations would be very convincing one way or another.

As described above we have now carefully analysed the ratio of cells that have gained or lost a chromosome arm and our finding that these cells occur at similar frequencies (new Fig. 5c), which is also supported by similar experiments by Allen et al. using Cas9, indicates that reciprocal chromosome exchange, and not large deletions, are the primary driver of LOH in our system.

2) Another consequence of the occurrence of extensive LOH is they bring into question several sections of the paper. These are as follows:

a. The entire section entitled “Mutations caused by multiplexed editing are locally restricted” along with figure 3 is rendered moot by the next sections that shows that extensive LOH is detected for many sgRNA arrays. Moreover, if the results in Figure 3 are reevaluated after acknowledging that chromosomal sequences distal from the centromere and towards the telomere are often deleted following DSBs induced with sgRNA arrays, it adequately explains why sgRNA arrays centromere proximal to *se* at *GstO3* affect its function but not at the telomere proximal *Ra* locus, when both arrays are practically equidistant from the beginning and end of *se* coding sequences respectively and why the telomere proximal array at *CG5892* has no effect on the centromere proximal *ebony* locus. Internal sgRNAs at *forked* showed no effect, but this may reflect that *forked* is essential to viability in fruit flies and therefore large deletions are lost. Therefore, this entire section should be

removed, and could potentially be replaced with a further analysis of the deletion alleles generated at the target sites by sgRNA arrays used in actual gene knock-out studies, similar to the analysis in Figure 1C.

We respectfully disagree with this assessment. The experiments in Figure 3 (now revised Figure 4) directly address a critical practical question: whether our system generates false positives due to disruption of neighboring genes. We acknowledge that the original analysis of four gene pairs was limited in scope and have substantially expanded this investigation.

We selected additional gene pairs where the telomere-proximal gene has a well-characterized, strong knockout phenotype, while the centromere-proximal gene has an unknown phenotype. The revised Figure 4 now presents 10 additional comparisons, each involving a gene essential for wing morphogenesis. Targeting these reference genes with HD12aCFD sgRNA arrays produces strong wing size reductions, indicating highly efficient knockout. Critically, targeting closely neighboring genes does not affect wing morphogenesis, demonstrating that proximity effects do not occur at practically relevant frequencies.

These findings align with our previous data targeting genes near pigmentation loci, which revealed proximity effects for only one of four sgRNA arrays. Notably, this array targets sites located towards the telomere and in extremely close proximity to *sepia*, located just tens to hundreds of base pairs from the transcriptional start site, a distance much closer than the vast majority of target sites of HD12aCFD arrays are to neighboring loci.

In addition, we now have also investigated two HD12aCFD sgRNA arrays targeting near haploinsufficient loci (new Suppl. Fig. 4). While rare in the genome, these loci are expected to be particularly sensitive to proximity effects. We observed no phenotype in one case and only mild phenotypes in the other, further supporting our conclusion that proximity effects are not a major driver of HD12aCFD phenotypes.

Collectively, these experiments demonstrate that phenotypes typically result from on-target gene disruption rather than proximity effects. The results with *GstO3* and one haploinsufficient locus also illustrate that in rare cases neighboring loci can be affected, giving rise to mild phenotypes, providing an important cautionary example that we now clearly highlight for users (Lines 609-617). The *Drosophila* community maintains rigorous quality control standards for genetic perturbation studies, including validation with orthogonal approaches, and is well-equipped to navigate such considerations.

b. The section that utilizes LOH to determine off-target effects of sgRNA arrays along with experiments in Figure 4 are unfortunately ill-conceived. The authors contend that a lack of off target LOH is evidence for lack of all off-target effects from sgRNA arrays. However, typical repair outcomes for Cas9 DSBs are mostly small indels, therefore off-target effects are also similarly expected to mostly induce small indels. And it is important to acknowledge that the off-target sites of the sgRNA arrays may not necessarily be clustered like the on-target site, therefore even if LOH outcomes are as frequent as localized indels on the sgRNA target sites, the same may not be true for off-target sites. Off-target sites also may not be reproducibly cleaved, and thus the non-reproducible LOH observed could also be from off-target cleavage instead of spontaneous events. To compare this to a recent study by Höijer et al. in Zebra fish (PMID: 35110541) that characterized off-target repair outcomes at predicted off-target sites, 76% were small deletions, 4% large deletions

(>50bps) and 25% were small insertions. Even allowing for differences in repair outcomes between fruit flies and zebra fish and for Cas9 and Cas12a DSBs, nevertheless the LOH assay alone cannot account for the presence or absence of small indels and therefore cannot eliminate the possibility of off-target effects. Unless the authors perform Illumina sequencing or other NGS on bulk DNA from flies that express a select few HD12aCFD arrays at multiple predicted off-target sites for all four individual sgRNAs, the LOH analysis alone does not eliminate the possibility of off-target effects.

We agree with the reviewer regarding the importance of acknowledging limitations in our off-target analysis, and we have revised the manuscript accordingly (Lines 411-412, 625-628). Like any detection method, LOH screening has inherent sensitivity limits, and off-target events occurring below this threshold could remain undetected. We now emphasize this caveat more explicitly in the revised text.

The reviewer also raises an important mechanistic question that is important to judge the suitability of our LOH assay for off-target detection: whether isolated DSBs caused by single sgRNAs (expected at potential off-target sites) also induce robust LOH. We have experimentally addressed this question by directly comparing LOH induction between a complete HD12aCFD array and a single sgRNA targeting the same locus (for two independent loci). As demonstrated in the new Suppl. Fig. 7, single sgRNAs also induce robust copy-number neutral LOH, establishing that isolated DSBs at off-target sites would be detectable using our screening approach. This result further supports the validity of our off-target detection method, which we believe is one of the most innovative aspects of our study. We note that systematic off-target data involving many sgRNAs has not been previously reported in *Drosophila*. Compared to our originally submitted manuscript, we have now doubled the proportion of the genome screened for nuclease activity (new Fig. 5j), substantially expanding the data available to evaluate on- and off-target activity with HD12aCFD arrays.

c. The final two sections entitled “Multiplexed Cas12a+ gene targeting outperforms established Cas9-based methods” and “Increased gene disruption efficiency reveals a somatic function for trade embargo” are also complicated by the potential for LOH. Data in Figure 5 will be greatly strengthened if LOH could also be assayed at any of the three genes targeted, if there is a genetically assayable marker telomere proximal to these respective genes. And the novel function of *trem* may also be a consequence of LOH mediated knock out of a gene telomere proximal to *trem* on 3R, but this is a difficult question to address. Unfortunately, the two characterized mutants of *trem* with identified indels both proved to be homozygous lethal, and thus the wing development function could not be corroborated. Nevertheless, if the experiments previously suggested show LOH frequency is comparable to local deletions for high efficiency sgRNAs, the assertions in Figure 5 need to be qualified by stating a lot of the cells with gene knock-outs potentially contain chromosome truncations. And those in Figure 6 regarding a novel wing development function for *trem* ought to be removed entirely, if the possibility of a confounding effect due to frequent LOH cannot be eliminated.

The experiments described above demonstrate that LOH in *Drosophila* is predominantly copy-number neutral and that phenotypes from HD12aCFD gene targeting are typically not caused by proximity effects. In addition, we have now obtained the two publicly available RNAi lines targeting *trem* and have analysed their effect on wing morphogenesis (see Reviewer Figure below). One of those lines produces qualitatively identical, though

quantitatively weaker, phenotypes to those observed with HD12aCFD arrays. Since RNAi acts post-transcriptionally without affecting genomic DNA, these phenotypes cannot result from deletions or chromosomal rearrangements. This orthogonal validation strongly supports that the observed phenotypes reflect genuine *trem* loss-of-function.

The *trem* experiments exemplify how the enhanced knockout efficiency of HD12aCFD arrays can reveal previously undetected gene functions, demonstrating the biological value of this approach. We have therefore retained these findings in the manuscript as an important proof-of-concept for the system's discovery potential.

Reviewer Figure: RNAi mediated knock-down of *trem* affects wing morphogenesis in *Drosophila*. Transgenic fly lines harboring inducible RNAi transgenes targeting *trem* were obtained from the BDSC (Line HMS02049 (a)) and VDRC (Line KK105113 (b)) stock centers. RNAi expression was induced with *pdm2-Gal4* in wing precursor cells and adult wing phenotypes were analyzed. While line HMS02049 had minimal effects, knock-down with line KK105113 resulted in loss of wing vein tissue (b, arrow heads) and moderate reduction in wing size. Note that while these phenotypes are weaker than those resulting from Cas12a+ mediated gene targeting with a HD12aCFD array (Fig.7), they are qualitatively the same. As RNAi does not act on DNA, this rules out artefacts arising from the induction of DNA breaks as causative for these phenotypes and instead suggests a specific role of *trem* in wing morphogenesis.

To reiterate, the major concerns regarding this manuscript involve experiments that address the frequency of LOH and its potentially confounding effects on other assertions made in the manuscript, without which publication cannot be recommended. To summarize in a sentence, if the increase in gene knockouts from HDCas12aCFD is primarily due to an increase in extensive LOH, then this basically nullifies the utility of this approach compared to conventional approaches that use one or two DSBs, instead of 4 clustered DSBs. Finally, a minor concern accompanying this is that manuscript itself ought to be re-written, to acknowledge the potentially confounding affects LOH can have for gene knock-out studies. Also, the reference in the legend for Figure 4 should refer to extended data Figure 3. Not 2.

We thank the reviewer for raising this critical concern. Given extensive data from other organisms, the hypothesis that multiplexed double-strand breaks could induce frequent large deletions and proximity effects was indeed well-founded and merited thorough investigation. The additional experiments we performed during revision have now definitively addressed this possibility, demonstrating that induced LOH in our system is predominantly copy-number neutral rather than deletion-based and does not typically confound experimental results. These findings reflect established differences in chromosome biology between *Drosophila* and other model systems. We have incorporated this mechanistic insight into the

revised manuscript to better contextualize our results within the broader literature. Ultimately, the reviewer's feedback has helped us to substantially strengthen our manuscript and make it better accessible to the broad audience of *Nature Communications*.

Reviewer #2 (Remarks to the Author):

In this manuscript by Port et al., the authors describe a highly efficient gene-targeting system in *Drosophila* that leverages multiplexed Cas12a sgRNAs. By utilizing quadruple sgRNA arrays alongside an enhanced Cas12a variant (Cas12a+), they demonstrate the system's ability to efficiently mutate targeted genes. The authors suggest that this approach offers improved knockout efficiency compared to traditional Cas9-based methods, revealing phenotypes that previous techniques may have missed.

The strength of this study is that it expands the CRISPR toolbox in *Drosophila* to Cas12a and describes effective strategies for constructing Cas12a/sgRNA reagents for tissue-specific gene editing. Using this strategy, the authors generated a small library of HD12aCFD sgRNA arrays targeting 241 genes, which can be a useful resource for the *Drosophila* community. In addition, the authors designed *in vivo* assays to test several other properties associated with CRISPR mutators that have not been commonly examined in *Drosophila*, including DSB-induced toxicity, DNA disruptions beyond the target sites, and off-target effects (even though that was limited to a specific chromosomal segment).

We thank the reviewer for their positive feedback. We are pleased to report that we have performed a major extension of both our study and the associated transgenic resource during revision. Our sgRNA library now encompasses lines targeting over 800 genes, which are now publicly available from the Vienna *Drosophila* Resource Center. We have also doubled the scope of our off-target assessment and now report data covering 33% of the *Drosophila* genome.

Although most of the experiments were nicely done and the results were mostly convincing (with the caveat of missing some technical information that is important for evaluation), the reviewer thinks that the authors have overstated the advantage of the Cas12a system over Cas9 systems. The authors' claims were based on unfair comparisons involving Cas9 sgRNAs with suboptimal designs and unequal numbers of sgRNAs for the two systems. Some caveats were presented as being unique to Cas9 systems but are in fact common to all CRISPR systems. Thus, the Cas12a system appears to be oversold while the strengths of the Cas9 system were understated.

The reviewer raises important questions about the relative advantages of the Cas12a-based system described here relative to currently used Cas9-based systems. In response, we have performed extensive additional experiments, including two large-scale *in vivo* screens, and clarifications throughout the text. We believe these revisions fully address the reviewer's concerns and provide a more complete assessment of our systems performance relative to current technology. We have also carefully revised the manuscript to avoid overstatements about the advantages of the HD12aCFD system. We detail our specific responses below.

Major concerns

1. A major conclusion of this study was that the Cas12a+ and HD12aCFD system is superior to Cas9 systems in gene mutagenesis (“gene targeting with Cas12a+ and multiplexed sgRNAs achieves superior gene disruption efficiency compared to established Cas9-based systems across diverse tissues”). However, this very general conclusion was based on incomplete/biased comparisons. The efficiency of CRISPR mutagenesis is affected by multiple factors, including the choice of targeting sites, gRNA scaffold sequence, the number of gRNAs used, and Cas protein expression level and temporal patterns. In the comparison of Cas9 and Cas12a systems in this study, even if the Cas9 and Cas12a have comparable expression (of which the technical information is missing), the influence of gRNA target sites, numbers, and gRNA scaffold were not sufficiently considered. Numerous studies have shown that the gRNA target sequences have a large impact on editing efficiency (e.g. DOI: 10.1038/nmeth.3543; DOI: 10.1038/nbt.3437), but the manuscript did not discuss what criteria were used to select the target sequences for either system and what measures were taken to ensure that they are efficient. More importantly, studies have shown that optimizing the gRNA scaffold sequence for Cas9 can dramatically increase Cas9 activity (DOI: 10.1016/j.cell.2013.12.001; DOI: 10.1126/science.aao0932). Further work in *Drosophila* has incorporated these optimizations and demonstrated their improved efficiencies over the original scaffold (DOI: 10.1534/genetics.118.301736; DOI: 10.1073/pnas.2014255118). However, the authors have completely ignored these improvements in this manuscript, and the Cas9 sgRNAs examined in this study seem to still use the original suboptimal scaffold. Thus, although the results in this study suggest that the new HD12aCFD system is better than the authors’ own HD_CFD system, they cannot be generalized for claiming that the Cas12a+ system is better than Cas9 systems. If this is the authors’ intention, they should compare HD12aCFD to similar Cas9 systems with the optimized gRNA scaffold and to make sure that the efficiency and the number of target sites for each gene tested are comparable for both systems.

This is a fair point. *In vivo* screens are time and labor intensive, so for practical reasons we had initially focused our comparisons on the HD_CFD Cas9 library that was readily available in our lab. However, other relevant Cas9-based resources for gene knockout in *Drosophila* differ in construct design and could therefore have different knockout efficiencies. Furthermore, additional improvements to Cas9 sgRNA design have recently been described. The reviewer is therefore correct to point out that the improved efficiency of the HD12aCFD system might be restricted to a subset of Cas9 systems. We have now addressed this limitation through additional experiments.

Most importantly, we have now compared our system to another major public resource for CRISPR knockout mutagenesis, which differs in important design parameters from the HD_CFD library. We ordered all available BDSC sgRNA lines targeting genes for which we also had HD12aCFD lines and performed side-by-side gene editing followed by quantitative assessment of wing size. The results presented in the new Figure 6g show that HD12aCFD lines identify phenotypes not recovered with lines obtained from the BDSC, but not vice versa. This represents a major extension of our previous findings and demonstrates that our newly described multiplexed sgRNA system also outperforms Cas9 systems with different designs.

We agree with the reviewer that CRISPR mutagenesis efficiency can be strongly influenced by sgRNA design parameters, and that an ideal comparison would use matched

designs. However, Cas12a and Cas9 use fundamentally different sgRNA scaffolds and cannot utilize sgRNAs with the same spacer sequences (due to different PAMs), preventing systematic testing of matched designs. While this means individual comparisons might reflect variations in spacer efficiency, such differences would average out over many comparisons. The fact that we performed over 250 pairwise comparisons across our two screens and consistently observe cases where HD12aCFD arrays outperform Cas9 sgRNA lines, but not the reverse, strongly suggests a systematic advantage of this resource over current technology. We agree it is important to clarify that these experiments inform which currently available resources are best suited for knockout mutagenesis, rather than which Cas nuclease can in principle perform this task best. We have added a new paragraph to the Discussion to make this distinction (Lines 568-581).

We also tested whether the improved Cas9 sgRNA vector suggested by the reviewer (described in PMID 33782117) could enhance the activity of existing sgRNA lines to match HD12aCFD arrays. Importantly, comparing two different Cas9 sgRNA vectors allows for a matched design. We cloned the same sgRNA spacers used in the reference HD_CFD lines into the improved pAc-U63-gRNA2.1 vector and inserted them into the same genomic landing site. Any differences in activity must therefore result from the plasmid design rather than target site identity or expression level. We performed these comparisons on three central regulators of Hh, Wnt, and Notch signaling, genes for which the true loss of function phenotype is known. While sgRNA expression from the improved vector resulted in a mild increase in phenotypic strength compared to the HD_CFD vector, HD12aCFD decisively outperformed both designs for all three target genes (new Suppl. Fig. 10).

We decided not to perform a comparison of Cas9 and Cas12a lines using the same number of sgRNAs in the arrays. While protocols for constructing larger Cas9 sgRNA arrays exist, they have a crucial practical disadvantage in that construction at scale is very cost and labor intensive. A recent 4x Cas9 sgRNA library for the human genome required 168,000 individual oligonucleotides, over 125,000 individual PCRs, and over 40,000 GoldenGate reactions to construct 42,000 plasmids (see methods in PMID 39633028). The authors of this work explicitly state that library construction took two full-time staff members 21 weeks. In comparison, a library of similar size of 4x Cas12a sgRNAs can be generated using 42,000 pool-synthesized oligonucleotides (which are much cheaper than individual oligos) and a single large-scale PCR and GoldenGate reaction. A task that can be completed by a single researcher in one week. Comparing 4x Cas12a and 4x Cas9 constructs would therefore suggest a false equivalence between these approaches, when in reality one is practical to create at scale while the other is not. We have incorporated these points into the revised Discussion.

Together, our new findings demonstrate that the multiplexed HD12aCFD sgRNA arrays also outperform other Cas9 sgRNA libraries and designs. We regret not having provided sufficient technical detail about the steps we took to ensure comparable sgRNA expression and design (same landing sites, same sgRNA design software) in the original submission and have addressed this in the revised manuscript.

2. The authors claimed that “Cas12a+ activity can be effectively controlled in time and space, as phenotypic changes remained confined to the targeted regions - a precision that has been challenging to achieve with Cas9”. However, this point was not convincingly shown. The Cas12a+

expression in this study was controlled and determined by the Gal4 lines used, not by some intrinsic differences between Cas12+ and Cas9. Similar Gal4/UAS strategies can be used to control the tissue-specificity of Cas9 as well. To support this claim, the authors need to present data demonstrating that Cas12a+ is intrinsically more specific than Cas9.

We agree with the reviewer that such a statement should be supported by more systematic and quantitative comparisons. As this is not a central point of our study we have opted to remove this section and plan to perform a comprehensive characterization of this parameter in the future.

3. To show that gene disruptions caused by HD12aCFD sgRNA arrays are locally restricted, the authors checked phenotypes associated with several recessive genes (Figure 3). However, to see the recessive phenotypes, both alleles of the tested genes need to be mutated. Thus, the lack of a phenotype does not necessarily correlate with the lack of DNA disruption, and the results do not truly reflect the frequency of broader disruptions at the targeted loci.

Thank you for pointing this out. Haploinsufficient loci are rare in the *Drosophila* genome (PMID 22445104), so focusing on proximity effects for recessive genes represents the most practically relevant scenario. We have substantially expanded this analysis in the revised manuscript (revised Fig. 4). However, we have now also analyzed proximity effects near two Minute loci, which encode haploinsufficient ribosomal protein genes (new Supplementary Fig. 4). Disruption of just one copy of these genes is known to result in cells being lost through cell competition. We find no evidence for such effects when targeting one gene in proximity to a Minute locus, and only mild apoptosis when targeting the second. The latter observation is consistent with disruption of neighboring loci at low frequency, although such phenotypes could also reflect fitness effects of the target gene itself, which remains uncharacterized. Overall, our expanded analysis of proximity effects shows that these are not major drivers of phenotypes observed with HD12aCFD arrays, but can occur in isolated instances. We have taken care to present a nuanced discussion of these findings and their implications in the revised manuscript.

4. In somatic targeting of the STAR reporter, if the editing outcomes in the majority of the cells are full deletions of the sequences between the first and last targeting sites, one would expect to see a relatively distinct PCR band corresponding to such deletions (because shorter amplicons are preferred products in genomic PCR). However, Figure 1b does not show this. The results instead suggest that full deletions are rare or at least low in frequency. The germline targeting results in Figure 1c cannot be used to infer somatic targeting, as DNA repair can differ significantly between the soma and the germline. While I agree that the results suggest that deletions (that can be between any two target sites) become more frequent with more target sites overall, the writing (especially by describing the results of somatic and germline data in the same sentence) can leave people a false impression about the frequency of large deletions in the soma.

Thank you for pointing this out. We have modified the text to provide a more nuanced description of our results. We now write that the smear observed in Fig. 1b indicates greater target locus diversification (resulting in more diverse heteroduplex amplicons). Deletions are

likely part of this diversification, but the reviewer is correct that it is not possible to make any statements about their abundance from such gels and that differences in deletion frequencies are likely to exist between different target loci and tissues.

5. In Figure 1a, how would the editing outcome be different if only the most distal target sites were available? In other words, how would the distance between two target sites affect the editing outcome?

That is an interesting question, but would require generating a new set of differently spaced STAR reporters, which we consider to be beyond the scope of this manuscript. However, the spacing of target sites in endogenous genes is naturally variable and while there are more confounding factors that complicate comparisons of mutation outcomes, the results presented in Fig 1d point towards increased target locus diversification also on these variably spaced target sites.

6. If the LOH events in the screening were truly due to background stochastic mitotic recombination, they could be observable even in the control experiment where Cas12a or sgRNAs were not expressed. Was this the case?

We have modified the text to provide a more balanced discussion of the three most likely explanations for the rare and isolated LOH events: screening errors, background LOH and low level off-target cutting. Obtaining definitive proof would be difficult for the first, require screening of thousands of discs for the second, and high coverage trio whole genome sequencing for the last, and is therefore beyond the scope of this manuscript. We think it is best to alert readers to all possibilities and in particular highlight that these could still be caused by off-target effects that occur so rarely that they were not reproducible in the sample size of our rescreen (now Line 411). While such off-targets would in most cases be too rare to significantly confound results, it is nevertheless important for users of this and similar technology to account for the possibility by using appropriate controls and validation strategies.

7. Some important technical information is missing for some experiments. It is difficult to evaluate the results without the information. Below are some examples.

a. In Figures 1d, 3, and 5, where were Cas9 and Cas12a expressed? What measures were taken to ensure that their expression levels were comparable? How were gRNAs expressed (by UAS or a ubiquitous promoter)? What was done to ensure that the gRNAs were expressed at comparable levels?

b. In Figure 5, how was inducible mutagenesis of Notch, neur, and smo done?

c. In Figure 1c, how many sequences are presented for each genotype/batch? How decisions were made regarding which sequences are included in the figure panel?

d. The manuscript did not talk about how gRNA target sites were selected and how to make sure that they would be efficient.

e. Scoring of the LOH index was not clearly defined.

We apologize for not always providing sufficient detail. We have now included these details in the figure legends and methods section of the revised manuscript and would like to thank the reviewer for their feedback.

Minor concerns

1. “CRISPR editing often produces genetics mosaics”. This sentence reads as if genetic mosaicism is avoidable. But in tissue-specific CRISPR experiments, DNA DSBs are generated and repaired independently in each cell in the manipulated tissue. The outcome should always be genetic mosaics.

Agreed. We have revised this sentence.

2. The authors claimed that “...for Cas9, the most widely used CRISPR nuclease, creating multiplexed arrays of more than two sgRNAs remains technically challenging due to their large size and repetitive nature.” It’s true that gRNAs for Cas9 are larger than those for Cas12a, but multiplexed Cas12a gRNAs also contain repetitive sequences. Several studies in recent years (including those published by some of the same authors) have already reported convenient methods for making multiplexed (e.g. 2-6) sgRNAs for Cas9 (REF). So, I am not entirely convinced by the authors’ argument that the Cas12a system is much better than Cas9 in this regard.

The crucial difference is that a quadruple Cas12a sgRNA array is small enough to be encoded on a single synthetic oligonucleotide, while an array with the same number of Cas9 sgRNAs is too large. This allows Cas12a arrays to be produced and cloned in pools, whereas Cas9 sgRNA arrays must be manually assembled. This makes cloning 4x Cas12a libraries orders of magnitude cheaper than 4x Cas9 sgRNA libraries. Please also see our response above, which illustrates this contrast with a concrete example.

3. Not all results in Supplementary Fig. 5 were described in the main text. Although some details were provided in the figure legend, the results should be at least briefly described in the main text.

Thank you for pointing this out. This has been corrected.

4. Supplementary tables were not provided.

There appears to have been a problem during the upload of the initial submission. We now provide all tables.

Reviewer #3 (Remarks to the Author):

Reviewer #4 (Remarks to the Author):

Port et.al. present a Cas12a-based multiplexed system that targets each gene with four sgRNAs to enhance editing efficiency, using *Drosophila* as an in vivo model. Their findings demonstrate that the synergistic action of multiple sgRNAs leads to efficient deletion formation, significantly increasing the proportion of loss-of-function mutations. To rigorously evaluate off-target effects, they introduce a novel screening assay that enables real-time visualization of CRISPR-induced chromosomal alterations in living organisms and demonstrated no detectable off-target effects. Furthermore, they reveal the key roles of *trem* in wing development by using multiplexed Cas12a system. Overall, the study is intriguing and most of the data support their conclusion. My major concern is that the Cas12a multiplexed system in *Drosophila* has been described by the same group but there is no clear description that what the improvement of the system is and there is no data to support this system is outperform than previous one.

We would like to thank the reviewer for their valuable feedback. Our 2020 paper (PMID 32843348) represents a proof-of-principle study that shows that Cas12a⁺ can in principle be used for multiplexed gene editing in *Drosophila*. Many highly influential method papers were not the first ones describing an approach, but rather those that made it practically usable (e.g. Brand and Perrimon, 1993 (>9000 citations) or Bischof et al., 2007 (>1900 citations)). These studies rigorously evaluated the potential of a previously described technology and introduced refined tools to use it. Our current manuscript follows this tradition by providing novel mechanistic insights into the mode of action of multiplexed gene targeting, systematically evaluates its safety and efficiency, and introduces a large-scale sgRNA resource accessible to the community through the Vienna *Drosophila* Stock Center. Such data and resources had so far been missing and represent a requirement for broad community adoption.

In addition, we are convinced that the use of LOH screening to visualize CRISPR activity over entire chromosome arms is an innovation of broad interest to the genome editing community. We are not aware of previous work describing the unbiased assessment of CRISPR activity using thousands of sgRNAs over a very large proportion of the genome. Given that off-target propensity of sgRNAs is highly variable and bioinformatic prediction of off-target sites imperfect, the scale of the search space and sgRNA number that can be tested with this technology are key advantages that will be of broad interest to the community.

Specific comments:

1. Figure 1a illustrates the report structure targeted by the Cas12a system, where the target sites are identical. However, there is no description of the structure of the multiplexed sgRNAs array. Are the same sgRNAs used to target these identical sites, or is the array multiplexed with various sgRNAs? Including a diagram to depict the Cas12a sgRNAs system would be helpful. Additionally, if the array reporter could be designed to contain different target sites and be targeted by six distinct sgRNAs (with similar editing efficacy), it would provide further insights into assessing editing efficacy.

The reporter described in Fig. 1a-c uses multiplexing of the same target site that is targeted by a single sgRNA. The advantage of this system is that each site is targeted with the same probability, which allows us to assess how mutagenesis patterns evolve if additional cut sites are added. When targeting endogenous genes the efficacy of individual sgRNAs and their spacing is highly variable, which substantially complicates such analysis. We directly show this complexity in Figure 1d, which we believe is more relevant than generating a reporter with diverse cut sites. We have revised the text to more clearly describe the reporter system and the rationale behind it.

2. What is the x-axis in Figure 1c? How can the non-linear changes in deletion percentage among arrays be explained? For instance, why does the 2x array show a higher percentage of deletion than the 3x array in the first region? Additionally, while the 4x array exhibits the highest deletion percentage, there is a decline in the 5x and 6x arrays. Clarification on these trends would be helpful.

Figure 1c shows schematics of different germline transmitted alleles of the STAR reporter. The X axis represents the sequence of the reporter. Each line represents a sequencing read from an animal that has been individually genotyped, a laborious process that limits throughput of this assay. We believe this data clearly shows the potential to induce deletions between multiplexed target sites and also the heterogeneity of outcomes that stems from the fact the different repair modes compete with each other. However, we think to quantitatively compare the relative abundance of different outcomes between different designs would require many more datapoints, which would be practically difficult to achieve. We have edited the text and figure to clarify these points.

3. In Figure 1d, the authors compared the Cas12a multiplexed system with the Cas9 system. To draw a solid conclusion that Cas12a outperforms Cas9, it would be helpful to include the deletion efficacy of each single guide used in Figure 1d for both Cas12a and Cas9. As well as showing the deletion efficacy of Cas12a with two guides. Also, in figure 5 and 6.

While in principle such comparisons can be performed, as showcased in Fig. 1c, they are extremely laborious when performed over many conditions at the appropriate scale. As the mutation profile alone does not inform about the efficiency of gene knock-out (as multiple potential mechanisms of biological compensation exist), we have instead focused our efforts on answering a related question that is of higher practical relevance: Does multiplexed Cas12a gene targeting result in efficient and specific ablation of gene function? We believe our revised manuscript clearly shows that this is the case.

4. Although, the claim 'the use of multiple sgRNAs creates functional redundancy' has been demonstrated by previous studies, the data in Figure 1 does not provide enough support for this claim. The authors could strengthen this point by citing relevant literature or including additional data, such as single sgRNAs efficacy controls, to better support their argument.

We have revised the text and included additional citations to better support this argument.

5. Please explain the reason that the mutation test in figure 3 only use up to 2 guides instead of 4 guides.

We have performed a major extension of these experiments and present them now in the completely revised Figure 4. All of the described experiments used HD12aCFD arrays, which encode 4 sgRNA per gene.

6. The authors claim that off-target effects of Cas12a editing are undetectable. Is this due to the limitations of the detection system? Whole-genome sequencing would be useful to determine whether low-frequency off-target effects are present. Do these low off-target effects are universal in high order model or only in fly model. For future applications, in vivo off-targets information in organoid or mice model are useful (organoid or mice).

Like any other off-target detection method LOH screening will have a detection limit, which is set by the ratio of LOH and mutations at the cut site and the number of cells that are analysed in a given experiment. The data we present in Suppl. Fig. 6 shows that the sensitivity of the assay makes it well suited to detect off-targets that occur with practically relevant frequencies (it is important to note that what constitutes such 'practically relevant frequencies' will be very different when considering research in a model organism (the focus of this manuscript) or for example medical applications of CRISPR technology). While whole-genome sequencing is indeed a powerful way to detect off-targets, it is technically very challenging to distinguish natural genetic variation from off-target mutations in WGS data. As a result, informative experiments of that kind are rare and constitute entire manuscripts on their own. Moreover, WGS in combination with many sgRNAs, a crucial factor given the heterogeneity of off-target potential, is cost prohibitive. We therefore consider the use of WGS beyond the scope of this manuscript. While performing off-target analysis in mice or organoids is not the topic of our current study, we do provide references that highlight that LOH screening is also feasible in mammalian models.

7. Statistics is missing in figures 6b, 6c, and 6d.

Thank you for highlighting this omission, which we have now corrected.

NCOMMS-25-02242A: Point-by-point response – final revisions

[Author responses are in bold]

Reviewer #1 (Remarks to the Author):

Following on from the initial review of the manuscript, the primary concern raised by this reviewer was the possibility of chromosome-arm scale deletions induced by clustered Cas12a DSBs leading to copy number-loss Loss of Heterozygosity (CL-LOH), which would produce confounding proximity based phenotypic effects and limit the efficacy of this approach as a novel gene knockout strategy. Upon reading the revised manuscript, this reviewer commends the authors for embracing this concern and going above and beyond to address it. The experiments in the revised manuscript thoroughly demonstrating an absence of proximity based phenotypic effects from gene knockouts, and the presence of copy number gain events strongly support a predominance of DSB induced CN-LOH and a relative absence of DSB induced CL-LOH events. Moreover, the authors also produced a supporting figure showing that RNAi mediated trem knockdown also shows a similar, albeit weaker wing phenotype as the HDCas12aCFD mediated trem knockout, which ameliorate concerns regarding the novel proposed function of trem. This reviewer therefore recommends publication, notwithstanding a few minor comments that can be addressed with a few small changes in the manuscript text, which are as follows:

We would like to thank the reviewer for their kind words and their constructive and helpful feedback throughout the review process.

1) The copy-number neutral LOH (CN-LOH) events in the paper are ubiquitously described as mitotic recombination events. While it is true that allele specific Cas9 DSBs induce substantial levels of mitotic recombination associated with crossing over in *Drosophila* (PMID: 33750946), an important difference is that DSBs induced in this paper produce breaks on both homologues. This raises the formal possibility that CN-LOH events instead occur via reciprocal end-joining events, which have been demonstrated in human HT1080 cells, a p53-positive fibrosarcoma cell line (PMID: 32873648). The reviewer does acknowledge that frequent mitotic recombination has been demonstrated before in *Drosophila*, nevertheless acknowledging the possibility of reciprocal end-joining events covers all bases.

Thank you for this excellent point. We have adopted the broader term “interhomolog recombination” in lines 260, 453, 460 and now mention the possibility of reciprocal end joining in line 452 and the legend of Fig. 5a.

2) The method of tracking LOH on chromosome arms with integrated fluorescent reporters to detect off-target genome editing is indeed very convenient, but this approach is limited to tracking events only on the marked chromosome arms. This caveat is usually not acknowledged in the text, rather the phrase “off-target activity” is used without any qualifiers. It is therefore recommended that the text in lines 91 and 420 be changed to “off-target activity on the two chromosome arms with LOH reporters”.

We qualify the scope of our off-target assessment in lines 291, 292 (“any array-encoded sgRNAs with off-target activity on the screened chromosome arm are expected to lead to reproducible LOH”), lines 302, 303 (“In summary, LOH visualization enabled us to screen for Cas12a+ activity with over 2000 sgRNAs across 33% of the *Drosophila* genome.), and lines 470-472 (“Screening for nuclease activity across 33% of the *Drosophila* genome in

combination with over 2000 sgRNAs failed to detect any reproducible cases of off-target activity.”).

3) Figure 3 legend line 244 refers to supplementary figure 1, when it seems it should refer to supplementary figure 2.

Thank you. This has now been corrected.

Reviewer #2 (Remarks to the Author):

The authors have done a very nice job revising the manuscript by adding substantial new results. These new results and revisions of text addressed my previous concerns. The revised manuscript is a very nice piece of work which represents a significant advancement in CRISPR technology and application in *Drosophila*. I am supportive of its publication by Nature Communications but with two minor comments.

We would like to thank the reviewer for their encouragement and valuable feedback that has led to a significant improvement of our study.

1. Figure 2 characterized Cas12a+ activity using the GRACE reporter. UAS-Cas12a+ appears to have low levels of leaky activity in the wing disc. How is it compared to the leaky activities of previous UAS-Cas9 strains generated by the same group? This information would be useful for people deciding between the Cas9 and Cas12a+ system.

We appreciate this comment. While we have anecdotal observations suggesting that Cas12a+ exhibits reduced leaky activity compared to some of our previous Cas9 lines, a rigorous comparison would require testing multiple target sites, as differences in background activity could reflect variations in sgRNA efficiency rather than inherent differences between nucleases. We plan to address this in future work. The current manuscript focuses on demonstrating the substantial improvements in knockout efficiency achieved with the HD12aCFD system and establishing the high specificity of Cas12a+-mediated mutagenesis.

2. It is not clear how the sequencing results in Supplementary Figure 6 were generated. Were they derived from PCR from a single animal or a mixture of multiple larvae? Or were they from dissected larval wing discs?

We have clarified the experimental procedure in the figure legend of Suppl. Fig.6.

Reviewer #3 (Remarks to the Author):

Reviewer #4 (Remarks to the Author):

The authors have addressed most of my comments. I don't have any further questions.

Thank you for your valuable feedback throughout the review process.